# The contribution of transport emissions to ozone mixing ratios and methane lifetime in 2015 and 2050 in the Shared Socioeconomic Pathways (SSPs)

Mariano Mertens[1], Sabine Brinkop[1], Phoebe Graf[1], Volker Grewe[1,2], Johannes Hendricks[1],
Patrick Jöckel[1], Anna Lanteri[1], Sigrun Matthes[1], Vanessa S. Rieger[1,*], Mattia Righi[1], and Robin N. Thor[1]

[1]Deutsches Zentrum für Luft- und Raumfahrt, Institut für Physik der Atmosphäre, Oberpfaffenhofen, Germany
[2]Faculty of Aerospace Engineering, Delft University of Technology, 2629HS, Delft, the Netherlands
[*]Now at: Department of Meteorology and Geophysics, University of Vienna, Vienna, Austria

**Correspondence:** Mariano Mertens (mariano.mertens@dlr.de)

**Abstract.** We quantify the contributions of emissions from the transport sector to tropospheric ozone and the hydroxyl radical (OH) by means of model simulations with a global chemistry-climate model equipped with a source attribution method. For the first time we applied a method which allows to quantify also contributions to OH which is invariant upon disaggregation or recombination and additive. Based on these quantified contributions, we analyse the ozone radiative forcings (RF) and methane lifetime reductions attributable to emissions from the transport sectors. The contributions were analysed for each transport sector separately and for 2015 as well as for 2050 under the Shared Socioeconomic Pathways (SSPs) SSP1-1.9, SSP2-4.5, and SSP3-7.0. In line with previous publications using the source attribution approach, we quantify an ozone RF attributable to emissions from land transport, shipping, and aviation for the year 2015 of 121 mW m$^{-2}$, 60 mW m$^{-2}$, and 31 mW m$^{-2}$, respectively. At the same time, we diagnose a relative reduction in methane lifetime due to transport emissions of 14.3 % (land transport), 8.5 % (shipping), and 3.8 % (aviation). These reductions are significantly larger than reported by previous studies due to the application of the source attribution method. Compared to 2015, only SSP1-1.9 shows a strong decrease in ozone RF and methane lifetime reduction attributable to the entire transport sector in 2050. For the projections SSP2-4.5, we find similar effects of the total transport sector as for 2015, while the effects in SSP3-7.0 increase compared to 2015. This small change of the effects for the two projections compared to 2015 is caused by two main factors. Firstly, aviation emissions are projected to increase in SSP2-4.5 (increase of 107 %) and SSP3-7.0 (+86 %) compared to 2015, resulting in a projected ozone RF of 55 mW m$^{-2}$ (+78 %) and 50 mW m$^{-2}$ (+61 %) for the year 2050 from aviation emissions. Secondly, the non-linear effects of atmospheric chemistry in polluted regions such as Europe and North America lead to rather small reductions in ozone and OH in response to emission reductions, especially from land transport emissions. In addition, the increase in emissions from land transport in other parts of the world, particularly in South Asia, leads to an increased contribution of ozone and OH. In particular, ozone formed by land transport emissions from South Asia causes a strong RF that partially offsets the reductions in Europe and North America. Moreover, our results show that besides the non-linear response, lack of international co-operations, as in the SSP3-7.0 projection, hinder mitigation of ground-level ozone.

# 1  Introduction

Emissions from the transport sector, i.e., land transport, shipping, and aviation, affect climate and air quality. The climate is influenced by emissions of both, the long-lived greenhouse gas carbon dioxide ($CO_2$), and short-lived climate forcers such as nitrogen oxides ($NO_x = NO + NO_2$) or particles (Szopa et al., 2021). In contrast to the effects of the $CO_2$ emissions on the climate, non-$CO_2$ effects heavily depend on the atmospheric conditions at their emission point and their further transport and chemical transformation in the atmosphere. Examples for such complex processes are the formation of tropospheric ozone or secondary aerosols (Seinfeld and Pandis, 2006) and aerosol-cloud interactions (Bellouin et al., 2020). Tropospheric ozone is a greenhouse gas, which also affects air quality and is harmful to humans and vegetation (Monks et al., 2015; Fleming et al., 2018; Mills et al., 2018). Moreover, tropospheric ozone levels are closely linked to the tropospheric oxidation capacity, which determines the lifetime of methane, which also affects climate (e.g. Stevenson et al., 2013).

Tropospheric ozone is formed by photochemical reactions from precursor emissions such as $NO_x$, carbon monoxide (CO), methane ($CH_4$) and Volatile Organic Compounds (VOC). The formation of tropospheric ozone depends on the meteorological conditions and on the ratio of $NO_x$ to VOC concentrations in a non-linear way (e.g. Sillman, 1995; Archibald et al., 2020). Typically, two ozone formation regimes are distinguished: the so-called $NO_x$-limited regime, in which ozone production increases with increasing $NO_x$ concentrations, and the VOC-limited regime, in which ozone production increases with increasing VOC-concentrations but decreases with increasing $NO_x$-concentrations. Due to this non-linear behaviour, a relatively large reduction in precursor emissions will often only result in a relatively small reduction in ozone concentration. Depending on the prevailing chemical regime, emission reductions can even lead to an increase of $O_3$ in specific regions. Examples of this phenomenon could be observed during the emission reductions as a consequence of the countermeasures against the COVID-19 pandemic: although $NO_x$ emissions were greatly reduced, measurements and results of model simulations showed only a slight decrease in $O_3$ or, in some cases, regionally increasing $O_3$ values (e.g. Mertens et al., 2021; Matthias et al., 2021; Grange et al., 2021; Putero et al., 2023).

The emissions of the transport sector are an important source of ozone precursors and other species affecting climate and air quality. Due to various efforts in reducing the effect of the transport sector on climate and air quality, for example shifts towards electric vehicles, the emissions of the transport sector will likely undergo large changes in the future. When designing such mitigation measures for the transport sector these non-linear processes in atmospheric chemistry need to be considered. In this study, we therefore analyse the contributions of emissions from the three transport sectors – land transport (including road, rail and inland-shipping), aviation, and shipping – to the mixing ratios of ozone and the hydroxyl radical (OH) abundance in the troposphere. We analyse the present-day (2015) conditions and three future projections for 2050 and focus on non-linear changes of ozone and OH caused by emission changes. The year 2050 is chosen because it reflects a time horizon over which large changes of the transport sector are plausible, without exceedingly large speculations into the far future. For the future projections we selected three scenarios among the Shared Socioeconomic Pathways (SSPs Riahi et al., 2017): a 'green' scenario (SSP1; van Vuuren et al., 2017), a middle-of-the-road scenario (SSP2; Fricko et al., 2017) and a scenario of regional rivalry with low mitigative capacities (SSP3; Fujimori et al., 2017). We choose these scenarios as they span a broad range of possible

future emission developments. In addition, the differences of the emission totals are large enough to yield different chemical regimes. Hence, we will not focus on the details of the socioeconomic backgrounds and technological pathways behind the chosen scenarios. Rather, we will use the scenarios as a tool to study the chemical effects of transport emissions in different emission scenarios and to derive possible consequences for mitigation strategies. Here, we want to focus on three scientific questions:

- How large are the contributions of transport emissions to tropospheric ozone and what is their effect on climate?

- How do transport emissions influence the atmospheric oxidation capacity and with this the methane lifetime?

- How do these effects change under different future emission scenarios?

For the analyses, we apply the chemistry-climate model ECHAM/MESSy Atmospheric Chemistry (EMAC) which is equipped with a tagging technique for source attribution of ozone and ozone precursors including $HO_x$ (Grewe et al., 2017; Rieger et al., 2018). We performed simulations for 2015 and for the three considered SSPs. Each simulation covers five years and simulates the same present-day meteorology. Accordingly, the influence of climate change on atmospheric composition is not considered, but the effect on the results are discussed in detail.

This manuscript adds to the findings of Mertens et al. (2018). In the aforementioned article, a source appointment method, considering $NO_x$ and VOC at the same time, was applied for the first time to quantify the contribution of land transport and shipping emissions to tropospheric ozone. Here, we expand on those findings by adding the following important updates:

- We use a more recent emissions inventory including also consistent projections of the emissions for 2050 according to CMIP6.

- We integrate the aviation sector to compare the effects of all transport sectors in a consistent way using the same methods.

- We analyse the effects from land transport emissions of different regions.

- We apply a new tagging approach of OH and the hydroperoxyl radical $HO_2$ (Rieger et al., 2018) to also account for the contributions of transport emissions to OH and their effects on the methane lifetime.

Accordingly, the manuscript presents, to our knowledge, the first consistent analyses of the effects of transport emissions on tropospheric ozone using a source attribution approach considering all transport sectors and the effects on OH and the methane lifetime.

In this manuscript, we first describe the model system, the model set-up, the applied source attribution method in detail, and provide a short model evaluation in Sect. 2, followed by a summary of the transport emissions in the SSPs in Sect. 3. Subsequently, we discuss the contribution of the emissions from the three transport sectors on tropospheric ozone (in Sect. 4) and the transport-related ozone radiative forcings (Sect. 5). In Sect. 6, we investigate the contributions of the transport emissions to OH and to the methane lifetime. Finally, limitations of the study are discussed in Sect. 7.

## 2 Model description, numerical experiments and evaluation

In the following sections, the EMAC model, the source attribution model and the numerical experiments will be described in detail. In addition, we will discuss the transport emissions of the three considered SSP narratives.

### 2.1 The chemistry-climate model EMAC

For the present study, we applied the EMAC chemistry-climate model (Jöckel et al., 2006, 2010, 2016). EMAC uses the second version of the Modular Earth Submodel System (MESSy2) to link multi-institutional computer codes. The core atmospheric model is the 5th generation European Centre Hamburg general circulation model (ECHAM5, Roeckner et al., 2006). We utilized EMAC (ECHAM5 version 5.3.02, MESSy version d2.54.0.3-pre2.55-02-2077) in the T42L90MA-resolution, i.e., with a spherical truncation of T42 (corresponding to a quadratic Gaussian grid of approx. 2.8 by 2.8 degrees in latitude and longitude) with 90 vertical hybrid pressure levels up to 0.01 hPa.

The simulation set-up is very similar to the simulation *RC1SD-base-10a* described in detail by Jöckel et al. (2016) alongside with an evaluation of the resulting model simulations. Therefore, we describe here only the most important details of the set-up and differences compared to *RC1SD-base-10a*:

– The applied gas phase mechanism in the submodel MECCA (Module Efficiently Calculating the Chemistry of the Atmosphere, Sander et al., 2011) incorporates the chemistry of ozone, methane, and odd nitrogen. Alkanes and alkenes are considered up to $C_4$, while the oxidation of isoprene ($C_5H_8$) and some non-methane hydrocarbons (NMHCs) are described with the Mainz Isopren Mechanism, version 1 (von Kuhlmann et al., 2004). The mechanisms of MECCA is part of the Supplement.

– Heterogeneous reactions in the stratosphere (submodel MSBM, Jöckel et al., 2010) as well as aqueous phase chemistry and scavenging (submodel SCAV, Tost et al., 2006) are included. Photolysis rates are calculated using JVAL (Sander et al., 2014).

– Emissions of methane ($CH_4$), the long lived greenhouse-gases (e.g. $CO_2$, $N_2O$, etc.) and the ozone depleting substances (ODS) are not considered explicitly. Instead, pseudo-emissions of these emissions are calculated using the submodel TNUDGE (Kerkweg et al., 2006). TNUDGE relaxes mixing ratios in the lowest model layer towards prescribed mixing ratios using Newtonian relaxation (see also Jöckel et al., 2016).

– Dry deposition is considered via the submodel DDEP (described as DRYDEP by Kerkweg et al., 2006). It is based on the big-leaf approach by Wesely and Hicks (2000).

– The emissions of lightning $NO_x$ depend on the meteorology and are parameterised after Grewe et al. (2001), resulting in global total emissions of $\approx 4.5\ \mathrm{Tg(N)\,a^{-1}}$. This value is well in the range of the best estimate given by Schumann and Huntrieser (2007) that is based on a review of observational and modelling approaches.

- Emissions of $NO_x$ from soil and biogenic $C_5H_8$ emissions also depend on the meteorology and are calculated using the MESSy submodel ONEMIS (Kerkweg et al., 2006), following the parametrisations by Yienger and Levy (1995) for soil-$NO_x$ and Guenther et al. (1995) for biogenic $C_5H_8$.

- The radiation is largely based on the original radiation scheme from ECHAM5 (Roeckner et al., 2003), but restructured and expanded with additional features such as multiple diagnostic calls as described by Dietmüller et al. (2016).

- The land surface model and the boundary layer implementation are modularized versions (see also Jöckel et al., 2016) of the original implementations of ECHAM5 described in detail by Roeckner et al. (2003).

## 2.2 Source attribution method

The TAGGING method has been applied in numerous studies to quantify the effects of the transport sectors on tropospheric ozone (Mertens et al., 2018, 2020a, b; Matthes et al., 2021; Mertens et al., 2021; Maruhashi et al., 2024). The concept of the source attribution methods (also known as tagging or labelling methods) fundamentally differs from the concept of perturbation methods (also known as zero-out, brute force etc., Wang et al., 2009; Grewe et al., 2010; Clappier et al., 2017; Mertens et al., 2020b).

In this study, ozone and its precursors are attributed to 18 source categories which are listed in Table 1. Most of these categories refer to specific emission sectors, but some others (for example production of ozone in the stratosphere) do not refer to such sectors. For land transport and anthropogenic non-traffic emissions we separate five different geographical source regions (see Sect. S8 in the Supplement for detailed definition). These are:

- Europe (EU)

- North America (NA)

- South Asia (SA)

- East Asia (EA)

- rest of the world (ROW)

For better readability, we denote absolute contributions as superscripted abbreviations. For example, $O_3^{SHP}$ and $OH^{SHP}$
denotes the absolute contributions of shipping emissions to $O_3$ and $OH$ mixing ratios, respectively. In addition to the abbreviations given in Table 1, we define the abbreviation *tra* indicating the contribution of land transport emissions from all regions (i.e., the sum of land transport from Europe, North America, East Asia, South Asia and the rest of the world). Relative contributions are defined as the mixing ratios of a specific category (e.g., $O_3^{SHP}$) divided by total mixing ratio of the species (e.g., $O_3$). Since the budget of the tagged species is closed, the relative contribution of the sum over all categories for each considered
species yields 100%. For example, the relative contribution of shipping emissions to ozone, denoted as $C^{SHP}(O_3)$, in percent, is defined as:

$$C^{\mathrm{SHP}}(O_3) = 100 \cdot \frac{O_3^{SHP}}{O_3}. \tag{1}$$

**Table 1.** Overview of the categories for the tagging source attribution applied in this study. Detailed definitions of the categories are given by Grewe et al. (2017). [1] The abbreviation IN stands for industry, but anthropogenic non traffic contains all non-traffic related activities i.e. industry, households, energy and agricultural including agricultural waste burning

| name | abbreviation | description |
|---|---|---|
| anthropogenic non-traffic Europe | IN[1]-EU | anthropogenic emissions not related to transport from Europe |
| anthropogenic non-traffic North America | IN-NA | anthropogenic emissions not related to transport from North America |
| anthropogenic non-traffic South Asia | IN-SA | anthropogenic emissions not related to transport from South Asia |
| anthropogenic non-traffic East Asia | IN-EA | anthropogenic emissions not related to transport from East Asia |
| anthropogenic non-traffic ROW | IN-ROW | anthropogenic emissions not related to transport from the rest of the world (ROW) |
| land transport Europe | LT-EU | land transport emissions from Europe |
| land transport North America | LT-NA | land transport emissions from North America |
| land transport South Asia | LT-SA | land transport emissions from South Asia |
| land transport East Asia | LT-EA | land transport emissions from East Asia |
| land transport ROW | LT-ROW | land transport (IPCC codes 1A3b_c_e) emissions from rest of the world |
| land transport total | TRA | The category is not tagged individually, but calculated as the sum of LT-EU, LT-NA, LT-SA, LT-EA and LT-ROW |
| shipping | SHP | Global shipping emissions (IPCC code 1A3d) |
| aviation | AIR | Global aviation emissions |
| stratosphere | STR | downward transport from the stratosphere (more specifically, production of ozone by $O_2$) |
| biogenic | SOIL | biogenic emissions and soil-$NO_x$ |
| biomass burning | BB | emissions from biomass burning |
| CH4 | CH4 | degradation of $CH_4$ as source of NMHCs |
| N2O | N2O | degradation of $N_2O$ as source of $NO_y$ |
| lightning | LIG | emissions of lightning $NO_x$ |

## 2.3 Radiative forcing calculations

Based on the diagnosed ozone contributions, we calculate the contribution of each category to the radiative flux (stratospherically adjusted) at the top of the atmosphere during model runtime using multiple diagnostic calls of the radiation routines (the submodel RAD, Dietmüller et al., 2016). In the following, we refer to this contribution to the radiative flux as the "radiative forcing (denoted as RF) of the respective emission category". As discussed in detail by Mertens et al. (2018) our definition of the RF does not take into account the change of ozone since pre-industrial times (see Forster et al., 2021), as we perform no pre-industrial simulation. Instead, the ozone RF in each simulation is decomposed into the respective contributions from the 18 source categories for present day/future conditions (see also Dahlmann et al., 2011). The change of ozone attributed to natural sources between pre-industrial and present-day caused by the increase of anthropogenic emissions and the respective effects on the ozone chemistry are not considered (see detailed discussion in the Supplement of Mertens et al., 2018).

A detailed description of the procedure to calculate the RF is given by Mertens et al. (2018). Here we recall only the most important aspects, exemplarily for the shipping (SHP) sector:

1. $\Delta O_3{}^{SHP} = O_3 - O_3^{SHP}$ is calculated;

2. Stratospheric adjusted radiative fluxes are calculated at every radiation time step (i.e. every third model time step) for $O_3$ and $\Delta O_3{}^{SHP}$.

3. In post-processing, the radiative contribution is calculated by subtracting the radiative fluxes (rflux) at the top of the atmosphere:

$$RF_{O3}^{SHP} = \texttt{rflux}(O_3) - \texttt{rflux}(\Delta O_3{}^{SHP}), \tag{2}$$

It is important to note that neither the radiative fluxes from $O_3$ nor the ozone contributions (e.g. $O_3^{SHP}$) feed back back onto the dynamics. Instead, prescribed climatologies are used for the forcing of the dynamics (as described in Sect. 2.4).

## 2.4 Numerical experiments and emissions

For the present study, four different numerical experiments were performed as summarised in Table 2. Each simulation has been conducted for the period $07/2012-12/2017$. The year 2012 is considered as spin-up and the years $2013-2017$ are analysed. The chemical tracers, the tagging tracers, and water vapour are initialised in all simulations from a 1.5-years-long spin-up simulation. This initialisation is the same in each simulation.

EMAC is nudged by Newtonian relaxation of temperature, divergence, vorticity and the logarithm of surface pressure (Jöckel et al., 2006) towards ERA-Interim reanalysis data (Dee et al., 2011) . Furthermore, the sea surface temperature and sea ice coverage are prescribed as transient time-series from ERA-Interim.

To allow for identical, even binary identical, meteorological conditions in all simulations, the Quasi Chemistry Transport Model mode (QCTM mode; Deckert et al., 2011) of EMAC is used. The use of QCTM mode enables us to quantify signals also

**Table 2.** Overview of the performed numerical simulations.

| name | emission scenario | emission year | meteorology |
|------|-------------------|---------------|-------------|
| PD | SSP2-4.5 | 2015 | 07/2012 – 12/2017 |
| SSP1-1.9 | SSP2-4.5 | 2050 | 07/2012 – 12/2017 |
| SSP2-4.5 | SSP1-1.9 | 2050 | 07/2012 – 12/2017 |
| SSP3-7.0 | SSP3-7.0 | 2050 | 07/2012 – 12/2017 |

for very small emissions perturbations, which otherwise (i.e. with a full coupling of chemistry and dynamics) would be hard to detect in a statistically significant way or would need very long integration times. In this mode, mixing ratios of the radiatively active trace gases are prescribed for the radiation calculations. This means that in each simulation the same radiative forcings by the prescribed mixing ratios are considered. Furthermore, mixing ratios of the chemical species are prescribed for the processes responsible for the coupling of the chemistry with the hydrological cycle. The prescribed mixing ratios are monthly averages taken from a previous model simulation performed with the *PD* setup, but with full coupling of chemistry and dynamics.

The simulations are performed as time slices, meaning that prescribed monthly emissions for 2015 or 2050 are repeated every year. For each simulation, the emissions of short-lived chemical species ($NO_x$, $NH_3$, $SO_2$, CO and VOC) from biomass burning, anthropogenic non-traffic sources, land transport, shipping, and aviation sectors are taken from the respective emission scenarios. The emissions of longer-lived species (including methane) and ozone depleting substances, which are nudged as described in Sect. 2.1, are taken from the SSP2-4.5 emission inventory for 2015. Emissions of soil $NO_x$, biogenic isoprene, and lightning $NO_x$ are calculated online, driven by the meteorological conditions, and are therefore identical in each simulation. The transport emissions in the simulations are discussed in detail in Sect. 3, including an analysis of the $NO_x$ emission totals. More information on the applied emissions (also for CO and VOC) are given in Sect. S1 in the Supplement.

## 2.5 Model evaluation

The EMAC model has been extensively evaluated in the past. Jöckel et al. (2016) present a detailed evaluation of various atmospheric variables, including tropospheric and stratospheric ozone. From these evaluations we know that EMAC has a positive bias of tropospheric ozone and a negative bias of carbon monoxide. Estimates of the methane lifetime simulated by EMAC are typically at the lower end of the range of values estimated by other models. However, multi-model inter-comparisons show that the biases compared to observational data of EMAC are within the range of those of comparable models (Naik et al., 2013; Stevenson et al., 2013; Voulgarakis et al., 2013; Young et al., 2013).

Given these extensive previous evaluation efforts, we reduce the evaluation of our model results to a minimum. In a first step we compare the ozone mixing ratios of the results from our PD simulation with the results of the *RC1SD-base-10a* simulation discussed by Jöckel et al. (2016). The set-up of both simulations are very similar, despite changes of the emission inventories, small updates and bug-fixes in the model infrastructure, and the fact that we simulate more recent years. Ozone is larger by $2-4$ nmol mol$^{-1}$ in PD compared to *RC1SD-base-10a* in the extra-tropical lower and middle troposphere. In the

extra-tropical free troposphere the difference between the two simulation is slightly larger, reaching up to 8 nmol mol$^{-1}$. In
the tropical troposphere the difference range between -2$-$2 nmol mol$^{-1}$. Overall, the change is lower than 8 % with the largest
increase in the Southern Hemisphere, dominated by the variability of the polar vortex. Figures of the comparison of ozone and
of further trace gases are provided in the Supplement (see Supplement Sect. S10). From this analysis we conclude that the
extensive evaluation presented by Jöckel et al. (2016) remains valid.

In addition, we compared the simulated ozone mixing ratios in the upper troposphere and lower stratosphere (UTLS) with
Satellite measurements published as the Stratospheric Water and OzOne Satellite Homogenized dataset (SWOOSH) by Davis
et al. (2016). The SWOOSH data are a homogenized, gridded, monthly-mean data set for ozone and water vapour based on
several satellite data. For the considered period the data set is based on Aura MLS. We used the SWOOSH data in version 2.6
with a horizontal resolution of 2.5° and 31 vertical levels. Horizontally, the SWOOSH data are interpolated onto the slightly
coarser EMAC grid, vertically the data are interpolated onto the much coarser SWOOSH grid similar as by Pletzer and Grewe
(2024). The monthly-mean SWOOSH data are compared with monthly-mean data from the model, meaning that satellite data
and model data are not co-located in space and time. Averaging Kernels of the Satellite are not considered, accordingly the
satellite data can only be used for a qualitative evaluation. The evaluation is performed for the years 2013$-$2017.

Figure A1 shows the difference between the ozone mixing ratios of the PD simulation and the SWOOSH data. Overall, the
inter-comparison confirms the known bias of simulated ozone, as discussed above, also in the upper troposphere. We would like
to stress that the results can only be used for qualitative evaluation (i.e. confirming the ozone bias), as neither averaging kernels
are used, nor are the data spatially and temporally co-located. Moreover, the number of considered years are very limited and
we found that the magnitude and location of the peak of the upper tropospheric ozone bias strongly depends on the approach
used for vertical remapping due to the limited vertical resolution of SWOOSH. For a detailed quantitative evaluation of UTLS
ozone we refer to previous inter-comparisons for example with the IAGOS (in-situ measurements on board passenger aircraft)
measurements presented by Jöckel et al. (2016); Pletzer et al. (2022); Cohen et al. (2024).

## 3 Transport emissions in the SSP projections

In our study, we consider three specific emission scenarios among the various combinations of SSP projections and forcing
pathways (Gidden et al., 2019):

- SSP1-1.9: A version of the SSP1 narrative with a target radiative forcing from anthropogenic emissions of 1.9 W m$^{-2}$
  in 2100;

- SSP2-4.5: A version of the SSP2 narrative with a target radiative forcing from anthropogenic emissions of 4.5 W m$^{-2}$
  in 2100;

- SSP3-7.0: A version of the SSP3 narrative with a target radiative forcing from anthropogenic emissions of 7.0 W m$^{-2}$
  in 2100.

The narratives of these the SSPs can be summarized as following (O'Neill et al., 2017):

- SSP1 (Sustainability): SSP1 features a relatively low population growth with a medium-high economic growth per capita. There is moderate international trade, effective international cooperation and policy focuses on a sustainable development. Technological development is rapid with a strong change away from fossil fuels and towards efficiency and renewable energy.

- SSP2 (Middle of the road): SSP2 features a medium population growth and a medium economical growth per capita. International cooperation is relatively weak and international trade is moderate. The environmental policy has a concern for local pollutants, but there is only a weak focus on sustainability in the policy orientation. Accordingly, there is only some investment in renewable energy and fossil fuel continues to play an important role. Technology developments is medium and distributed unevenly, with a slow transfer of technologies.

- SSP3 (Regional Rivalry): SSP3 features a high growth of the population in the group of high- and low-fertility countries (see KC and Lutz, 2017, for more details) and a low growth in rich OECD (Organization for Economic Co-operation and Development) countries. The economic growth per capita is slow and there is a high inequality across countries. There is de-globalisation and a strongly constrained international trade. International cooperation's are weak and there is a low priority for environmental issues. Technological developments are slow with a low technological change directed
towards domestic energy sources.

The traffic emissions of air pollutants between the three projections differ strongly, as they depend on the mobility, global trade, as well as on the applied technologies. Overall, in 2050, passenger travel demand is largest in SSP2, followed by SSP1 and SSP3. Cars are an important transportation mode in all three SSPs, with the largest passenger travel demand in SSP2, followed by SSP3 and SSP1. In SSP1, public transport (bus and train) plays a more important role than in SSP2 and
SSP3. Moreover, SSP1 features the lowest demand for aviation, followed by SSP3 and SSP2. High-speed trains become more important as an alternative mode of transportation in SSP1 (see Fig. 5 in van Vuuren et al., 2017). The SSP1 has the overall lowest energy demand for the transport sector, followed by SSP2 and SSP3. Even though the energy demand of the transport sector in SSP1 is lower than in SSP3, SSP1 has the larger transport demand. This shows that a decoupling of energy demand and passenger demand is achievable in a sustainable scenario. In all SSPs, liquid fuels dominate the overall energy demand for
transportation in all scenarios until ~2040. Only in the SSP1, electrification of cars increase compared to present day already before 2040. In this scenario, hydrogen also becomes an important energy carrier in the transport sector after 2050. In the SSP3 projection, even in 2050, only very little electrification is projected in the transport sector, while the SSP2 projection assumes a moderate amount of electrification (see Fig. S13 in the Supplement of Bauer et al., 2017). Generally, SSP1 features a maximum electrification of 75 % of total transport, while SSP2 and SSP3 feature 50 % and 10 % maximum electrification, respectively
(Fricko et al., 2017).

In addition to the adaptation of alternative energy sources, such as electricity, in the transport sector, air pollutant emissions depend heavily on the regulations that are enacted. Different pollution control levels have been defined which are linked with the SSP narratives (Rao et al., 2017):

- SSP1: Policy makers aim for much lower air pollution levels over the 21st century than the current legislation targets. There is a strong concern for negative effects of air pollutants on health and ecosystems. Moreover, medium and low income countries catch up relatively quickly with industrialized countries in terms of pollution control.

- SSP2: The general trend of emission reductions is continued worldwide and the pollution concentration targets become more ambitious with increasing income over the century. There is increasing concern for effects of pollutants on human health and the ecosystems. Due to diffusion of technologies and knowledge, developing countries reach levels of emission control relatively early compared to historical developments in OECD countries. Regions with large population densities may not reach the targeted air pollution levels.

- SSP3: Compared to SSP2, the implementation of pollution control is delayed. Moreover, due to less international cooperation and slow deployment of international laws, technologies are developed less quickly and trans-boundary pollution (e.g. hemispheric transport of air pollutants) is increasing.

The integrated assessment models (IAMs) behind the SSPs do not apply complex air pollution control technologies. Therefore, the pollution control scenarios are implemented in a simplified approach based on trajectories of the emission factors (Rao et al., 2017). The trajectories are based on two sets of emissions factors. The first set is „current legislation" (CLE), assuming efficient implementation of existing environmental legislation. The second set is the Maximum Technically Feasible Reduction (MFTR) set. The near-term achievement of MFTR is not considered feasible, but it serves as a baseline for what is ultimately possible given the current state of air pollution control technologies.

SSP2 assumes an implementation of CLE until 2030, while SSP1 (SSP3) assume a stronger (less strict) implementation of CLE until 2030, respectively. After 2030, the progress of the emission factors depends on the level of technological progress, i.e., the progress towards MFTR emission factors. In addition, depending on their economic developments, non-OECD countries can catch up with OECD countries with respect to the implementation of air pollution control faster (SSP1) or slower (SSP3). In all scenarios, the MFTR emission factors are stagnant in time, which means that there is no speculation about possible future technologies better than MFTR. Thus, in countries and SSPs with high penetration of MFTR, this might be a conservative scenario (Rao et al., 2017).

In all SSP projections, the spatial distribution of the land transport emissions is taken from the EDGAR (Emissions Database for Global Atmospheric Research, Janssens-Maenhout et al., 2019) 4.3.2 emission inventory, aviation emissions are distributed as in CMIP5 (Coupled Model Intercomparison Project, phase 5, Taylor et al., 2012) and the shipping emissions are distributed according to ECLIPSE (Evaluating the Climate and Air Quality Impacts of Short-Lived Pollutants, Stohl et al., 2015). The spatial distribution of land transport and aviation emissions are constant in time. Shipping emissions use projected changes of sulphur emissions due to changes in sulphur content and the definition of the Sulphur Emission Control Area (SECA). More details on the regridding process of the CMIP6 emissions are given by Feng et al. (2020). We apply the aviation emissions in their original form which have an error in the geographical distribution, as recently reported by Thor et al. (2023). The corrected emissions would lead to an increase of the ozone RF by 7.6 %.

Figure 1 displays the $NO_x$ emissions in the four simulations. Tables of the total emissions for $NO_x$, VOC and CO emissions are part of the Supplement (Sect. S1). Further analyses of the changes of the total emissions in the SSP projections are given by Righi et al. (2023).

In the SSP1-1.9 projection for 2050, anthropogenic $NO_x$ emissions reduce by more than 70 % compared to 2015, which is in agreement with the underlying narrative. Especially the emissions in the transport sector decrease strongly by around 90 % for most land transport regions, around 95 % for the shipping sector, and 64 % for the aviation sector. The reductions are caused by transformations of the transport system and by alternative fuels (e.g. electrification). Generally, SSP1 features a much larger share of public transport (e.g., buses and trains) in 2050 compared to the other SSPs (van Vuuren et al., 2017)

and the aviation sector has a lower passenger demand compared to SSP2 and SSP3. In addition to this high share of public transport and the quick shift towards alternative technologies, SSP1 also features the strongest air pollution control among the considered scenarios.

In the SSP2-4.5 projection for 2050, anthropogenic $NO_x$ emissions decrease by around 25 %, but emission peaks are shifted regionally compared to 2015. As an example, land transport emissions decrease in most regions, but strongly increase over

315 South Asia (around 60 %), and over Africa. The decrease is mainly caused by stronger air pollution control, especially in the OECD regions, while the increase is partly caused by increases of the population in South Asia and parts of Africa. The aviation emissions are doubled compared to 2015, which is consistent with the increase of passenger travel demand in the aviation sector. The shipping emissions are reduced by around 30 %. These reductions mainly take place around Europe and the East coast of North America, while emissions increase, especially along the main shipping routes from China to Singapore.

Most of the reductions are likely due to technological improvements (including increased air pollution control).

The SSP3-7.0 projection for 2050 exhibits around 12 % larger anthropogenic $NO_x$ emissions compared to 2015. The regional changes, however, differ strongly. For example, the land transport $NO_x$ emissions decrease in Europe and North America compared to 2015 by around 40 % and 50 % respectively, as a consequence of the stronger air pollution control (incl. technological advancements) in these regions. In South Asia, the $NO_x$ emissions from land transport increase by almost 100 % and

325 in the rest of the world by around 40 %. One reason for the particularly strong increase in South Asia is the strong growth of population, in particular in India, in SSP2 and SSP3 in 2050 compared to 2015 (KC and Lutz, 2017). In addition, air pollution control is limited in SSP3. The shipping emissions for 2050 are lower in SSP3 compared to SSP2, even though the air pollution control is less strict in SSP3 compared to SSP2. This is likely caused by the strong decrease of international trade between the regions in SSP3 (Fujimori et al., 2017; Bauer et al., 2017).

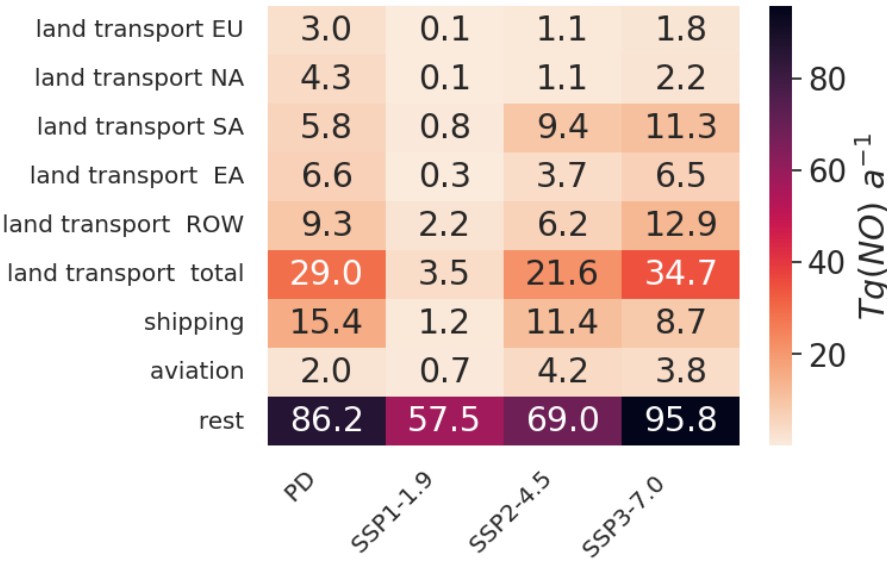

**Figure 1.** Comparison of the $NO_x$ emissions (in $Tg(NO)\ a^{-1}$) between the four performed simulations for 2015 (PD) and 2050 (projections, see Sect. 2.4 for a detailed description). The land transport emissions are given for the different source regions and as global totals ("land transport total"). 'rest' indicates all other emission sources, including also soil- and lightning-$NO_x$ emissions which are not included in the SSP emission data. Please note, that the "land transport total" emissions are only given for comparison, they are not emitted in addition to the regional land transport emissions.

## 4  Contributions to ground-level ozone

We first analyse the general changes in ground-level ozone for the three projections compared to 2015 (Sect. 4.1), and follow with an analysis of the individual transportation sectors in detail in Sect. 4.2 to Sect. 4.4. In these analyses the contributions to ground-level ozone (land transport and shipping) and zonally averaged contributions to ozone (aviation) are analysed. Moreover, we quantify the trends of the absolute contributions from land transport and shipping emissions to ground-level ozone for specific geographical regions. For the aviation sector we focus on the zonal means of the relative contributions and the trends of the contribution of aviation ozone to the tropospheric ozone column for main flight regions. Moreover, we quantify the changes of the ozone burden caused by the changes of the emissions in Sect. 4.5.

### 4.1  Changes of ground-level ozone

Figure 2 shows the ground-level ozone mixing ratios for 2015 and the changes of the ground-level ozone mixing ratios compared to 2015 for the three projections in 2050. Figures of the ground-level ozone mixing ratios for the projections are part of the Supplement (Sect. S2).

In 2050, ozone decreases in the SSP2-4.5 projection compared to 2015 in most regions by $2-6 \, \mathrm{nmol \, mol^{-1}}$. However, ozone increases slightly over Central Africa, around India, along the east coast of China and in Central Europe. Especially in Central Europe and China, the increase is due to decreased ozone titration caused by decreasing $NO_x$ emissions. For the SSP3-7.0 projection, ozone increases strongly in 2050 compared to 2015. The increase reaches up to $10 \, \mathrm{nmol \, mol^{-1}}$ and affects especially West Africa, the Arabian Peninsula, India, and South-East Asia. In contrast to this, ground-level ozone decreases strongly in 2050 for the SSP1-1.9 projection compared to 2015. In most regions the decrease is in the range of $10-15 \, \mathrm{nmol \, mol^{-1}}$, and exceeds $20 \, \mathrm{nmol \, mol^{-1}}$ on the Arabian Peninsula. The general changes of ground-level ozone for the three projections and regional features, such as the strong increase of ozone over Asia in SSP3-7.0, are in agreement with the analyses of CMIP6 simulation results by Turnock et al. (2020). However, the magnitude of the ozone change differs, especially our increase of ozone in SSP3.7-0 is lower as shown by Turnock et al. (2020). Moreover, our results for the SSP3-7.0 does not show the strong decrease of ozone over the oceans as discussed by Zanis et al. (2022). Both differences can be expected, as we keep the methane lower boundary condition to present-day values, and because we do not include the effects of changing meteorology and climate and therefore also have constant water vapour concentrations in all simulations (see Sect. 7 for a detailed discussion).

### 4.2  Land transport

Figure 3 shows the relative contributions of land transport emissions to ground-level ozone as simulated in the four experiments. In addition, the absolute contribution for 2015 and the changes in the absolute contribution for the various regions are shown as an average over specific regions in Fig. 4. These regions are: Europe, North America, East Asia, South Asia, Southeast Asia and South America (see Appendix Sect. S9.1 for a detailed definition). Additional figures depicting absolute contributions at ground-level and zonal means are part of the supplement (Sect. S3 and S4 therein).

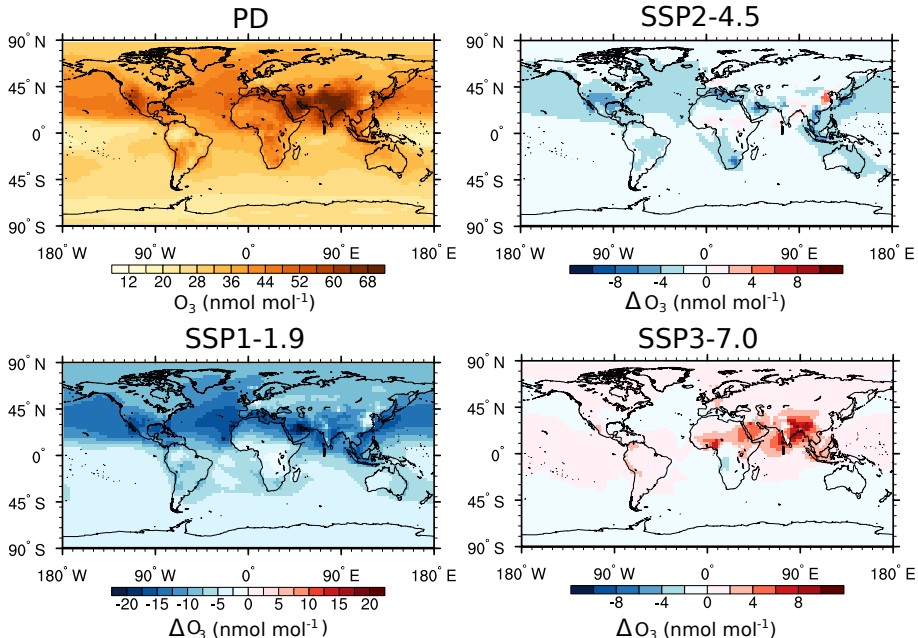

**Figure 2.** Ozone mixing ratios (5 year average) at ground-level for the PD simulation and differences ($\Delta O_3$, '2050 minus 2015') for SSP1-1.9, SSP2-4.5, and SSP3-7.0. Please note that SSP1-1.9 has a different color scale.

Generally, the absolute and relative contributions over North America, Europe, and the Middle East for present-day conditions (simulation PD) are comparable to previous studies (see Mertens et al., 2018, and detailed tables therein). The contributions for present-day conditions over India and parts of South-East Asia, however, are larger compared to previous studies. This increase is mainly caused by the larger emissions in these regions in the Community Emissions Data System (CEDS) emission inventory compared to other emission inventories (e.g. Hoesly et al., 2018; McDuffie et al., 2020). Moreover, a more recent emission year (2015) has been used here.

For the SSP2-4.5 projection in 2050, the relative contributions of emissions from land transport to ground-level ozone ($C^{\mathrm{tra}}(O_3)$) at the hotspots over India and Southeast Asia increase to 30 % and 25 % respectively (see Fig. 3). The absolute contributions increase over India, while they decrease over South East Asia (see Fig. 4), but due to the reduced ozone levels $C^{\mathrm{tra}}(O_3)$ increases. The increase over India is consistent with the increase of emissions from land transport in the SSP2 narrative, as population grows fast and air pollution control strategies and technological development are not strong enough to counteract the increased demand of mobility. Over Europe, East Asia, and North America, absolute and relative contributions of land transport emissions decrease due to stricter air pollution policies compared to present day.

The SSP1-1.9 emission pathways shows a strong reduction of land transport emissions in all regions of the world, which lead to a strong reduction of the contributions to ozone (see Fig. 3). The largest levels of $C^{\mathrm{tra}}(O_3)$ are simulated in the Middle East and South Asia (8−9 %). As $O_3$ itself decreases in the SSP1-1.9 (see Fig. 2), absolute contributions decrease at an ever

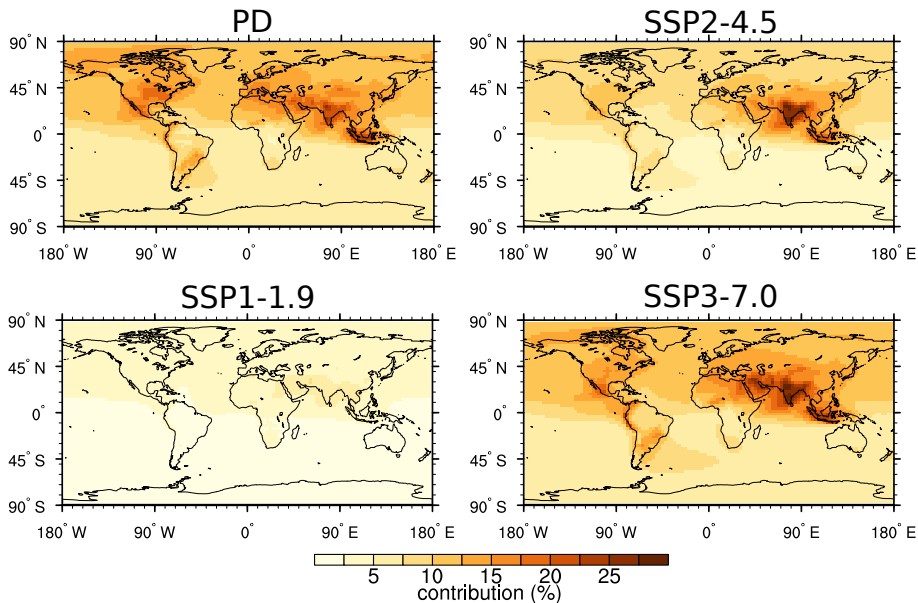

**Figure 3.** Relative contributions of land transport emissions to ground-level ozone (in %) for the simulations PD, SSP1-1.9, SSP2-4.5, and SSP3-7.0 (5 year average). The colour bars are identical in all panels to allow for a better comparability between the results for the different emission projections.

stronger pace (see Fig. 4). This strong decrease is caused by the strong change of the transport system towards public transport, and electrification of the transport system as well as the very strong air pollution mitigation efforts.

The SSP3-7.0 projection in 2050 features globally even larger land transport emissions compared to present-day, caused by the combination of population growth, slow technological development and less strict air pollution mitigation. This increase of the emissions globally leads to increasing contributions of the land transport emissions to ozone, but regional changes vary. Even though, land transport $NO_x$ emissions are reduced by around 50 % compared to 2015 over North America and Europe, relative (16−18 %) and absolute contributions decrease only slightly. A detailed analyses of the reason will follow in Sect. 4.5.

Over the Middle East, India, and South America $C^{tra}(O_3)$ increases up to 28 %, 31 % and 22 %, respectively. In these regions also absolute contributions increase. This increase is caused by an increase of emissions due to the combination of increased mobility demand, lack of public transport, and failure of effective emission control.

To quantify whether the changes in the contribution of land transport emissions to ground-level ozone for the future projections are caused by changes in regional emissions or by emission changes in other regions of the world, we performed a

source-receptor analysis (see Fig. 5 and Fig. S18 in the Supplement). We define the tagged regions (see Table 1) as sources, and the regions as used in Fig. 4 as receptor regions. Please note, that the region Southeast Asia is not defined as separate source regions, but listed under rest of the world (ROW).

For all receptor regions in 2015, regional land transport emissions are the most important source for the contribution of land transport emissions to ozone. This means, for example, that land transport emissions from the source region Europe contribute

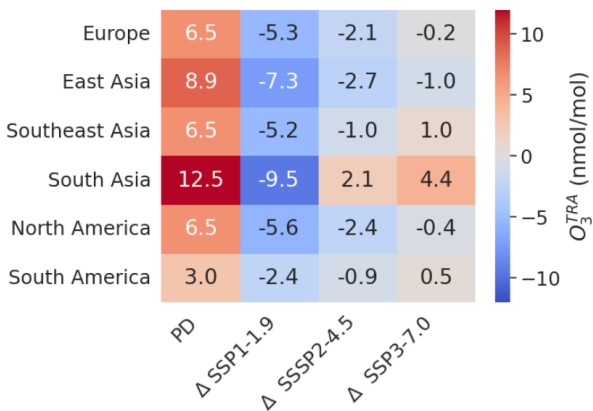

**Figure 4.** Absolute contributions of land transport emissions to ground-level ozone for 2015 and the differences of the absolute contributions for the three 2050 simulations ('2050 minus 2015', all in $\text{nmol mol}^{-1}$). Values are area averaged over specific geographical areas and for 5 years (see Sect. S9.1 in the Supplement for exact definition of the regions).

most to the contribution of land transport emissions to ozone in the receptor region Europe. This changes in the projections for 2050, particularly for Europe and North America. Here, the contribution of regional land transport emissions declines sharply due to air pollution control and the source regions of the ROW have (partly) larger contributions compared to regional emissions.

Interestingly, only the emissions of the source regions North America, South Asia, and East Asia have larger influence on the contributions of land transport emissions for other receptor regions. The contributions of European land transport emissions to ground level ozone are confined around Europe (see also additional Figures in Sect. S3 in the Supplement).

The large contributions of land transport emissions for present day and in the projections SSP2-4.5 and SSP3-7.0, especially for South Asia, shows the importance of strong mitigation measures in these regions to effectively reduce $O_3$ levels. SSP1-1.9 shows one possible mitigation pathway, leading to a strong reduction of land transport emissions and its contribution to ozone over South Asia by strong changes of the transport system and strong pollution control. In addition, the population is less strongly increasing, leading to a lower demand for mobility in SSP1-1.9 compared to SSP2-4.5 and SSP3-7.0 in South Asia.

Overall, the importance of regional emissions in most receptor regions indicates a large benefit for ground-level ozone when reducing regional emissions. However, the increasing importance of the contribution of land transport emissions from other regions in the projections shows how important it is to reduce ozone precursors not only regionally, but also globally. Especially the increase of the ROW contribution for Europe, South Asia and North America in SSP3-7.0, in which international cooperations are strongly reduced, shows how important these international co-operations are in order to reduce air pollution by long range transport.

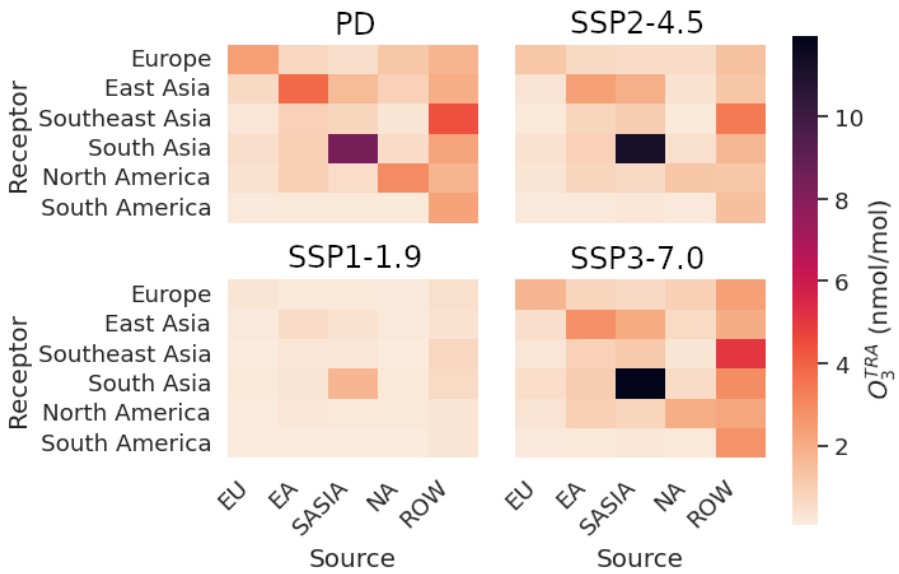

**Figure 5.** Source receptor analysis of the absolute contribution of land transport emissions to ground-level ozone (in $\mathrm{nmol\,mol^{-1}}$). The values are mean values over 5 years and area weighted over the receptor regions. Exact definitions of the receptor regions are given in Sect. S9.1 in the Supplement.

## 4.3 Shipping

Figure 6 shows the relative contributions of shipping emissions to ozone ($C^{ship}(O_3)$) at ground level. Quantifications of the absolute contribution of shipping emissions for specific geographical regions, which are Atlantic, Indian Ocean, Pacific and the Mediterranean Sea (see Sect. S9.2 in the Supplement for a detailed definition), are displayed in Fig. 7. Additional figures depicting absolute contributions at ground-level and zonal means are part of the Supplement (Sect. S3 and S4 therein).

Generally, $C^{SHP}(O_3)$ is largest in the North Eastern Pacific and the North Eastern Atlantic. In 2015, $C^{SHP}(O_3)$ reaches up to 18 % in these regions, corresponding to mixing ratios of $O_3^{SHP}$ of up to 7 $\mathrm{nmol\,mol^{-1}}$ in the North Eastern Pacific and up to 12 $\mathrm{nmol\,mol^{-1}}$ along the Western Coast of North Africa. Similarly, to the land transport emissions, the geographical distribution of shipping emissions for present day is comparable to previous studies (Mertens et al., 2018; Butler et al., 2020). Despite higher emissions from the shipping sector, the simulated contributions in the present study are slightly lower compared to Mertens et al. (2018), which is probably due to the larger overall $NO_x$ emissions in the present study. Compared to Butler et al. (2020), the simulated contributions in the present study are lower. This is due to the differences in the tagging methods used. While Butler et al. (2020) performed either $NO_x$ or VOC tagging, we consider all precursor and their competition in ozone production an loss reactions simultaneously (see Sect. 7).

In contrast to the land transport sector, where the geographical patterns of the contributions change strongly due to the regional emission shifts resulting from the local nature of the mitigation measures in this sector, the geographical pattern of the regions with the largest contributions from the shipping sector remains very similar in 2050 for SSP2-4.5 and SSP3-7.0. This can be expected as also the shipping emissions show no strong regional shifts (besides the introduction of SECAs) and mitigation measures are mostly applied globally due to technological advancements by means of improved emission factors (i.e. technological advancements). The absolute and relative contributions decrease to around 16 % (8 $\mathrm{nmol\,mol^{-1}}$) for SSP2-4.5 and 14 % (7 $\mathrm{nmol\,mol^{-1}}$) for SSP3-7.0. The lower contributions of shipping emissions in SSP3-7.0 compared to SSP2-4.5 is in accordance with lower shipping emissions caused by the decrease of global trade in the SSP3 narrative (see Sect. 3). The strong decrease of shipping emissions in SSP1-1.9 leads to strong reduction of the absolute and relative contributions from shipping emissions to ozone. Accordingly, $C^{ship}(O_3)$ drops below 4 % and $O_3^{ship}$ to 2 $\mathrm{nmol\,mol^{-1}}$ at maximum. If climate-change would be considered in addition, the ozone contribution from shipping emissions could be reduced even more strongly in the future, given the likely reduction of ozone over the oceans due to increasing humidity (Zanis et al., 2022, see also disucssion in Sect. 7).

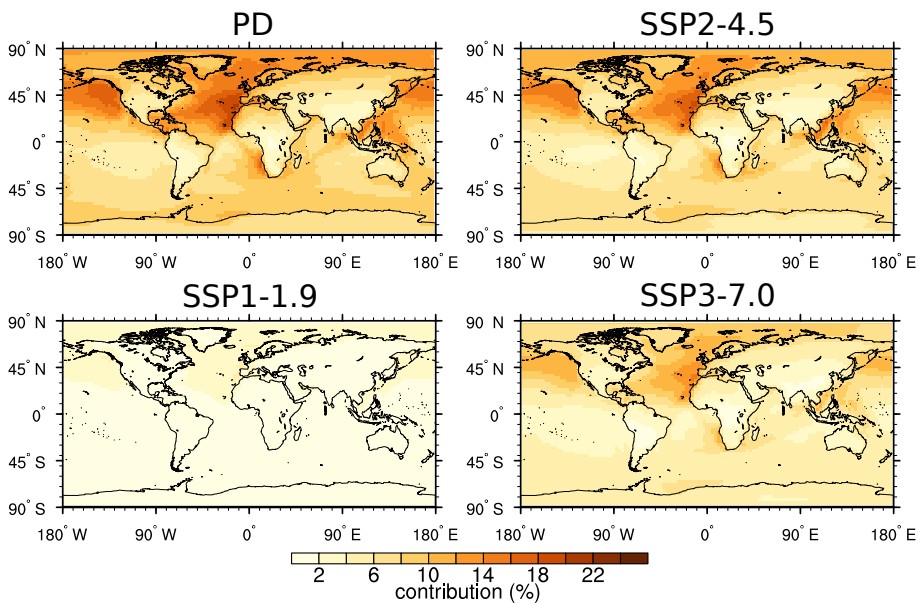

**Figure 6.** Relative contributions (in %) of shipping emissions to ground-level ozone as simulated by PD, SSP1-1.9, SSP2-4.5 and SSP3-7.0. All values are annual averages for 5 year average. The colour bars are identical in all panels to allow for a better comparability between the results for the different emission projections.

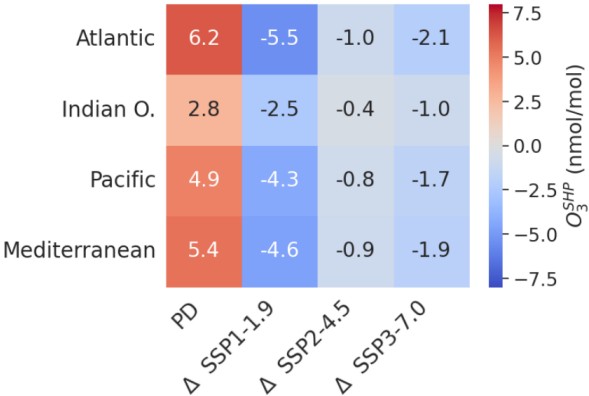

**Figure 7.** 5-year mean of the absolute contributions of shipping emissions to ground-level ozone for PD and the differences of the absolute contributions for the three 2050 simulations ('2050 minus 2015', all in $\mathrm{nmol\ mol^{-1}}$). Values are area-averaged for specific geographical regions (see Sect. S9.2 in the Supplement for a detailed definition).

## 4.4 Aviation

Figure 8 shows the zonal average of the relative contributions of aviation emissions to ozone ($C^{air}(O_3)$). Absolute contributions of aviation emissions to the tropospheric column of $O_3^{AIR}$ averaged over the main flight regions between North America and Asia, Asia and Europe and North America and Europe (see Sect. S9.3 in the Supplement for a detailed definition) are given in Fig. 9.

The largest values of the relative and absolute contributions in 2015 are simulated in the Northern hemisphere. They reach up to 5 % ($4-5$ nmol mol$^{-1}$) and occur over a large region between $60° - 90°N$ (see Fig. S10 in the Supplement). The largest relative contributions can be found between 800 and 300 hPa, while the largest absolute contributions are simulated around 250 hPa. Generally, the geographical distribution of the regions with the largest contributions are similar to the regions with the largest ozone changes as quantified with the perturbation approach reported by Koffi et al. (2010). The results of the perturbation approach, however, show a relative change of ozone which is much more limited to the upper troposphere. This is because below the cruise altitude ozone precursors from other sources are becoming more and more important. While in the perturbation approach the non-linearities in the net ozone production lead to a compensation of reduced emissions and increased net ozone production, here such an compensation is avoided.

The increase of the aviation emissions in SSP2-4.5 and SSP3-7.0 due to the increase of passenger travel demand lead to an increase of the ozone contributions. For the SSP2-4.5 projection, $C^{air}(O_3)$ reaches up to 10 % ($8-8.5$ nmol mol$^{-1}$) in the Northern hemisphere in 2050. Like for the shipping emissions, the aviation emissions are lower in the SSP3-7.0 emission pathway compared to the SSP2-4.5 pathway. Accordingly, maximum relative and absolute contributions of aviation emissions to ozone reach up to 8 % ($7-7.5$ nmol mol$^{-1}$) in SSP3-7.0 for 2050. Similar as the absolute and relative contributions, the tropospheric ozone columns of $O_3^{air}$ increase by almost 90 % (SSP2-4.5) and 70 % (SSP3-7.0) compared to present-day conditions (Fig. 9).

The SSP1-1.9 projection shows the lowest contributions in 2050 compared to the two other pathways, caused by decreased passenger travel demand compared to SSP2-4.5 and SSP3-7.0 and technological advancements compared to 2015. The maximum of $C^{air}(O_3)$ of around 4 % is only a slight reduction compared to the values in 2015. However, the background ozone is much lower in SSP1-1.9 for 2050 due to the overall decreased emissions, so that the 4 % relative contribution corresponds to maximum absolute contributions of 2.5 nmol mol$^{-1}$. Accordingly, also the tropospheric ozone column of $O_3^{air}$ decrease by around 50 % in SSP1-1.9 compared to present-day.

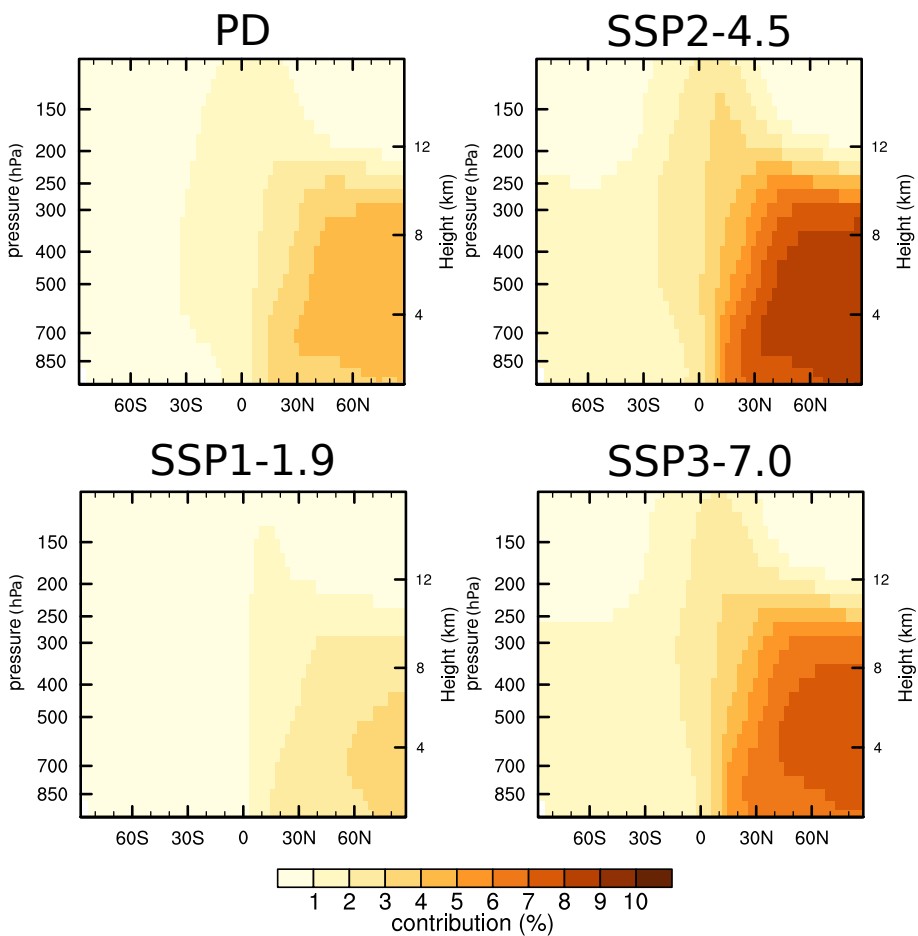

**Figure 8.** Zonal average of the relative contributions (in %) of aviation emissions to ozone as simulated by PD, SSP1-1.9, SSP2-4.5 and SSP3-7.0.All values are averages for 5 years. The colour bars are identical in all panels to allow for a better comparability between the results for the different emission projections.

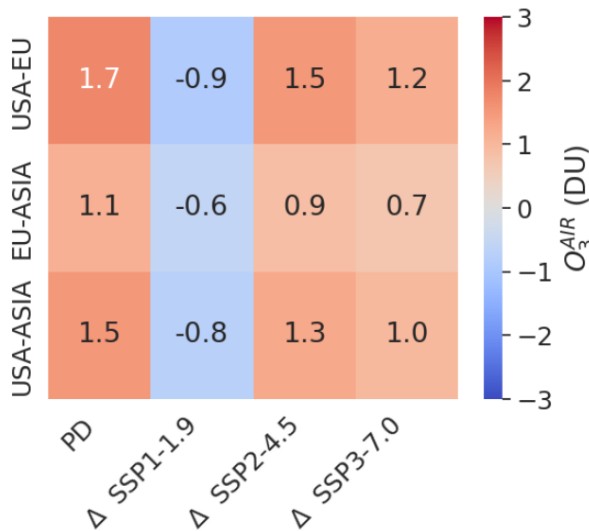

**Figure 9.** 5-year mean tropospheric $O_3^{AIR}$ column for *PD* (in Dobson Units, DU) and the differences of the tropospheric $O_3^{AIR}$ columns between the three 2050 simulations ('2050 minus 2015'). The values are area averaged over the the main flight regions (see Sect. S9.3 in the Supplement for a detailed definition).

## 4.5 Analysis of the non-linearity of the ozone chemistry

The production of tropospheric ozone is strongly non-linear and depends on the ratio of $NO_x$ to VOC concentrations (e.g., Sillman, 1995; Seinfeld and Pandis, 2006; Wu et al., 2009; Wild et al., 2012; Archibald et al., 2020). In the $NO_x$-limited regime, the ozone production efficiency (i.e. the amount of ozone molecules formed by one $NO_x$ molecule) increases if $NO_x$ emissions are reduced and decreases with an increase of the $NO_x$ emissions.

To analyse how the ozone burden in the atmosphere changes for the different projections, we introduce the following metric which we define exemplary for the shipping sector as following:

$$\chi^{SHP} = \frac{B(O_3^{SHP})}{E(NO_x^{SHP})}. \tag{3}$$

In this definition, $B(O_3^{SHP})$ is the global burden (in Tg) of $O_3^{SHP}$ and $E(NO_x^{SHP})$ are the annual $NO_x$ emissions of the shipping sector in Tg(NO). Increasing values of $\chi$ (called the ozone burden efficiency) indicate an increase of the ozone burden per emitted $NO_x$. We prefer this quantity over the commonly analysed ozone production efficiency as it includes production and loss processes of ozone and therefore is a better measure for the associated radiative forcings.

Figure 10 shows the dependency between $\chi$ and the $NO_x$ emissions. The results clearly show the expected increase of the ozone burden efficiency with decreasing emissions. Depending on the prevailing meteorological conditions and on the chemical background conditions, $\chi$ differs among the transport sectors and in different world regions (as analysed for the land transport emissions). Generally, $NO_x$ emissions from the aviation sector have a larger efficiency in forming ozone compared to surface emissions, because the emissions are emitted under cleaner background conditions. This is consistent with previous findings (e.g Hauglustaine et al., 1994; Derwent et al., 2001; Hoor et al., 2009; Dahlmann et al., 2011; Butler et al., 2020; Wang et al., 2022; Terrenoire et al., 2022). Land transport emissions in Europe or North America have rather low values of $\chi$ (at least) for present day, as they are emitted in a rather polluted background and the prevailing meteorological conditions do not favour ozone production over parts of North America and Europe. This is in agreement with the results of Zhang et al. (2016), showing a larger ozone production efficiency near tropical regions compared to mid-latitudes.

The increase of $\chi$ with decreasing $NO_x$ emissions underlines a major challenge for ozone mitigation. A very prominent example are the land transport emissions from Europe and North America in SSP3-7.0. Even though the $NO_x$ emissions are reduced by around 40 % (Europe) and 50 % (North America) compared to 2015, the global burdens of ozone produced from these emissions are reduced by 5 % (Europe) and 16 % (North America), only. Accordingly, reductions of $NO_x$ emissions from the land transport need to be large enough to overcome strong non-linear behaviour of the ozone chemistry so that the emission reductions lead to significant reductions in ozone. The SSP1-1.9 and SSP2-4.5 projections show two possible projections with such strong reductions over Europe and North America. As example, the reduction of $NO_x$ emissions by 63 % over Europe in SSP2-4.5 compared to 2015 leads to 30 % reduction of ozone from European land transport emissions.

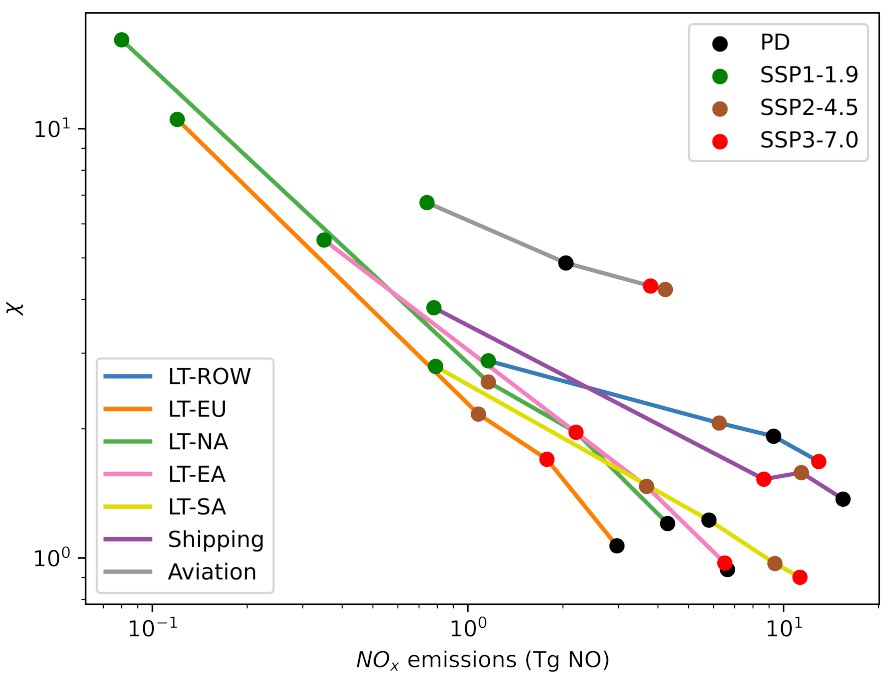

**Figure 10.** Dependency between $\chi$ and the annual $NO_x$ emissions (in Tg(NO)). $\chi$ is calculated as global mean over 5 years. Shown are the results of the 7 transport categories (land transport (indicated as LT-): ROW, EU, NA, SA and EA, aviation and shipping) for each simulation (colour coded).

## 5 Ozone radiative forcing

The stratospherically adjusted radiative forcings of ozone (for simplicity from now on called RFs) of the different transport sectors are depicted in Figure 11. RF values are given for the three main transport sectors (Fig. 11a) and for the land transport emissions from specific regions in Fig. 11b (see Table 1 for naming convention). The total RF of the land transport sector in 2015 is $121\,\mathrm{mW\,m^{-2}}$. This value is $29\,\mathrm{mW\,m^{-2}}$ ($\approx 32\,\%$) larger compared to Mertens et al. (2018), which is partly due to the $\sim 35\,\%$ larger $NO_x$ emissions from land transport in the present study. Moreover, especially the land transport emissions in Asia are larger in the present study, which have a rather high efficiency for RF (see Sect. 5.1).

The RF of the shipping sector in 2015 is $54\,\mathrm{mW\,m^{-2}}$, which is $8\,\mathrm{mW\,m^{-2}}$ ($\approx 13\,\%$) lower compared to Mertens et al. (2018), despite the about $20\,\%$ larger $NO_x$ emissions in the present study. This is most likely caused by the overall larger $NO_x$ emissions from anthropogenic sources in the present study compared to Mertens et al. (2018). This larger overall $NO_x$ emissions lead to a more polluted background and a lower ratio of the shipping emissions compared to all anthropogenic emissions (around $7\,\%$ in the present study compared to $12\,\%$).

The RF of the aviation sector in 2015 is $29\,\mathrm{mW\,m^{-2}}$, which lies between the best estimates by Lee et al. (2021) for 2011 ($27\,\mathrm{mW\,m^{-2}}$) and 2018 ($36\,\mathrm{mW\,m^{-2}}$). Note that the values from Lee et al. (2021) are based on a literature review from various models using the perturbation approach. Only a limited number of model results from EMAC are included in the multi-model assessment. The included results of EMAC tend to be at the lower end of the ozone response in Lee et al. (2021) and also in other multi-model assessments (Søvde et al., 2014, and Cohen et al., pers. communication). A recent assessment with EMAC, comparing different methodologies to asses the effects of aviation emissions on ozone, diagnoses a difference of ozone between the perturbation and tagging method of a factor of 1.16 to 2.55, with a mean value of 1.83 (Maruhashi et al., 2024).

As mentioned in Sect. 3, an inconsistency of the geographical distributions in the CMIP6 aviation emissions applied here, lead to an underestimation of the aviation $O_3$ RF of about $8\,\%$ (see Thor et al., 2023, for a detailed analysis).

The RF from land transport emissions is reduced to $103\,\mathrm{mW\,m^{-2}}$ in 2050 with the SSP2-4.5 projection. The strong increase of the emissions from the land transport sector in SA leads to an increase of the RF from $O_3^{\mathrm{LT-SA}}$ compared to 2015 of around $7\,\mathrm{mW\,m^{-2}}$. The RF of the land transport emissions of all other regions decrease slightly. The RF of the shipping emissions decreases slightly by $2\,\mathrm{mW\,m^{-2}}$ in 2050 compared to 2015, while the RF of the aviation sector almost doubles with SSP2-4.5. With this large increase, the RF of the aviation sector is slightly larger than the RF of the shipping sector in 2050 for SSP2-4.5. Moreover, the RF of the total transport sector (land, aviation and shipping) in 2050 with the SSP2-4.5 projection is almost as large as for present day conditions, as the increase of the RF in the aviation is almost of same magnitude as the reductions in shipping and land transport.

The strong increase of the land transport emissions in SSP3-7.0 compared to SSP2-4.5 leads to an increase of the RF from land transport emissions to $140\,\mathrm{mW\,m^{-2}}$ in 2050. In comparison to 2015 and to SSP2-4.5 (in 2050) especially the RF of the land transport emissions from the rest of the world (RoW) increases. Also the contributions of land transport emissions from South Asia increase mainly due to the strong increase of the emissions in India. The RF of the land transport emissions from

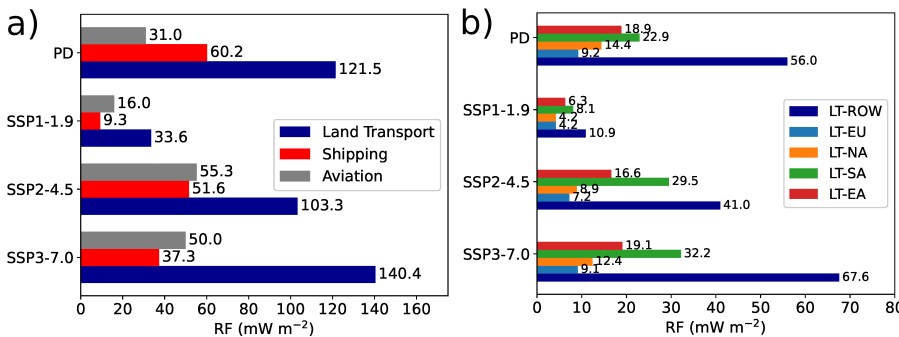

**Figure 11.** Stratospheric adjusted radiative forcing of ozone (in $\mathrm{mW\ m^{-2}}$) from each sector (a) and for land transport emissions from specific regions (b).

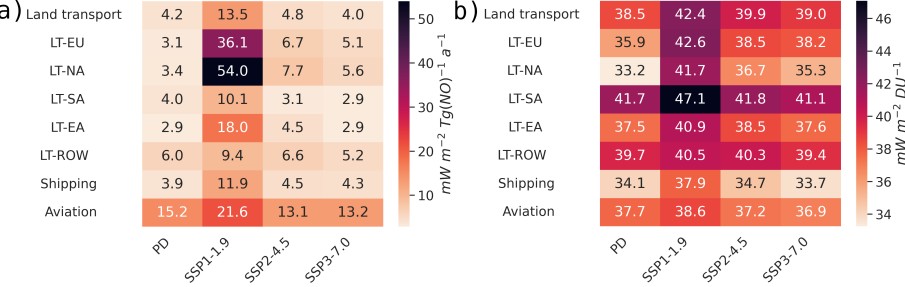

**Figure 12.** Global averages of $RF_e$ (a, in $\mathrm{mW\ m^{-2}((Tg(NO)\ a^{-1})^{-1})}$) and $RF_{DU}$ (b, in $\mathrm{mW\ m^{-2}DU^{-1}}$) for the emission sectors land transport, shipping and aviation as well as for the land transport emissions from specific regions only.

Europe, North America and East Asia stay relatively similar to 2015. Even though the $NO_x$ emissions from land transport are reduced strongly, ozone is only reduced slightly due to non-linear effects (see Sect. 4.5).

Consistent with the changes of $NO_x$ emissions and the $O_3$ contributions, RF from the aviation and shipping sector are smaller for SSP3-7.0 compared to SSP2-4.5. The RF from shipping emissions is $37\ \mathrm{mW\ m^{-2}}$ in 2050 with SSP3-7.0, which is a decrease of 17 and $15\ \mathrm{mW\ m^{-2}}$ compared to SSP2-4.5 in 2015 and 2050, respectively. The RF from aviation emissions 535   decreases for SSP3-7.0 by $5\ \mathrm{mW\ m^{-2}}$ compared to SSP2-4.5 in 2050, but increases by $21\ \mathrm{mW\ m^{-2}}$ compared to 2015.

In the SSP1-1.9 projection, RF from all transport sectors decreases strongly. The total land transport RF is reduced by a factor of four, the shipping RF by a factor of five and the RF from aviation emissions by a factor of three compared to 2015. Also the RF of the land transport emissions from various regions are reduced strongly compared to 2015 and also compared to the other projections.

## 5.1 Radiative Forcing efficiency

The RFs discussed in Sect. 5 are difficult to compare among the different sectors due to the different amount of emissions for each sector and region. To set the discussed RFs into perspective with the emissions of the individual emission sectors, we calculate the RF efficiency with respect to the $NO_x$ emissions for each sector (defined as $RF_e$, e.g., Dahlmann et al., 2011) as:

$$RF_e^i = \frac{RF^i}{E(NO_x^i)}, \qquad (4)$$

with $E(NO_x^i)$ denoting the $NO_x$ emissions in $Tg(NO)\ a^{-1}$ and $RF^i$ the stratospheric adjusted RF of the considered sector (denoted as i). In addition, we calculate the RF with respect to the tropospheric ozone columns, denoted as $O3_{tc}$, in Dobson Units (DU), for each emission sector (denoted with i) . We call this quantity $RF_{DU}$ and define it as:

$$RF_{DU}^i = \frac{RF^i}{O3_{tc}^i}. \qquad (5)$$

Figure 12 shows $RF_e$ and $RF_{DU}$ for the different projections and the different emission sectors. For present day, the aviation sector has the largest $RF_e$ value of $15.2\ mW\ m^{-2}\ Tg(NO)^{-1}\ a^{-1}$ ($32.5\ mW\ m^{-2}\ Tg(N)^{-1}\ a^{-1}$). This is in general in-line with previous studies (e.g Hauglustaine et al., 1994; Derwent et al., 2001; Hoor et al., 2009; Butler et al., 2020; Terrenoire et al., 2022) and the high ozone burden efficiency from aviation $NO_x$ emissions discussed in Sect. 4.5. Moreover, the value is $30\ \%$ ($7.5\ mW\ m^{-2}\ Tg(N)^{-1}\ a^{-1}$) larger as the multi-model mean of $25.1 \pm 7.3\ mW\ m^{-2}\ Tg(N)^{-1}\ a^{-1}$ by Lee et al. (2021) (see also discussion in Sect. 5). The inclusion of the correction of the CMIP6 aviation emissions as reported by Thor et al. (2023) increases the efficiency to $16.3\ mW\ m^{-2}\ Tg(NO)^{-1}\ a^{-1}$ ($34.9\ mW\ m^{-2}\ Tg(N)^{-1}\ a^{-1}$).

$RF_e$ for shipping is slightly smaller than for total land transport emissions, caused by the much more effective upward transport of land transport emissions by convection in the upper troposphere compared to shipping emission as already discussed by Hoor et al. (2009, see also zonal means in Sect. S4 in the Supplement). Land transport emissions from East Asia, Europe and North America show the smallest $RF_e$ values, because ozone is formed rather inefficiently from emissions of these regions (see Sect. 4.5). Moreover, especially for Europe and North America, the produced ozone is not transported into the upper troposphere very efficiently where it would be most radiatively active (e.g. Riese et al., 2012; Rap et al., 2015). The latter is analysed using $RF_{DU}$, which is largest for land transport emissions from South Asia. This is in agreement with results from Dahlmann et al. (2011) reporting a strong latitude dependence of $RF_{DU}$ with the highest values near the equator. Even though aviation has the largest $RF_e$ values, $RF_{DU}$ is lower as most emissions take place in mid-high latitudes (Dahlmann et al., 2011).

In the future projections, $RF_e$ increases for those emission sectors and regions whose emissions decrease, and $RF_e$ decreases where the emissions increase. The change of $RF_e$ agrees with the change of $\chi$. The large values of $RF_e$ for some sectors with SSP1-1.9 emissions are partly also dominated by the small amount of emissions, leading to numerical difficulties calculating the quotients.

The strong decrease of the emissions in the SSP1-1.9 projections lead to an increase of $RF_{DU}$, caused by reduced saturation effects of the radiative efficiency as analysed by Dahlmann et al. (2011). This increase of the radiative efficiency causes an additional penalty on ozone mitigation, as ozone gets more radiativly efficient if tropospheric ozone levels are strongly reduced.

## 6 Contributions of transport emissions to OH and to the methane lifetime

Emissions of the transport sector affect not only tropospheric ozone levels, but also have a strong influence on OH mixing ratios and with this on the tropospheric oxidation capacity. Increases in OH lead to a reduction of the methane lifetime. Previous studies analysed the influence of transport emissions on OH and methane lifetime with the perturbation approach (e.g., Hoor et al., 2009). Here, we determine the contribution of the emissions from the transport sector to OH and the associated contributions to the methane lifetime for the first time with a source attribution method. Due to the methodological differences, the results obtained in the present study are not directly comparable to previous studies. Therefore, we also performed three perturbation simulations, in which we applied a 5 % emission reduction to land transport, shipping and aviation, respectively. The changes of the methane lifetime by emissions from land transport, aviation and shipping with this method are in agreement with Hoor et al. (2009), who applied the same 5 % perturbation (see Sect. S5 in the Supplement).

### 6.1 Contributions to OH

Figure 13 depicts the zonal average of the contribution of land transport emissions to OH ($OH^{TRA}$). The zonal mean pattern of the contribution for the present-day simulation is similar as the one presented by Rieger et al. (2018). Two maxima of $OH^{tra}$ appear, one in the lower troposphere between the surface and 600 hPa at 0-30° N peaking at 20 fmol mol$^{-1}$, and one in the upper tropical troposphere between 100 and 200 hPa peaking at 26 fmol mol$^{-1}$. The maximum in the upper troposphere, however, is mainly due to the overall very large OH mixing ratio in these regions and the relative contributions in these regions are rather low (up to 5 % in 2015). Near the surface, the relative contributions of land transport emissions to OH reach up to 40 % in the NH in 2015. Hoor et al. (2009) reported only one maximum in the lower troposphere by applying the perturbation approach. This corroborates the fact that results of the attribution method and of the perturbation methods are not directly comparable. This second maximum of $OH^{TRA}$ is caused by photolysis of $O_3^{tra}$ in the upper troposphere and subsequent formation of OH.

Due to the strong interconnection between $NO_x$, $O_3$ and OH in atmospheric chemistry, the overall changes of $OH^{TRA}$ in the future projections are similar as discussed for the $O_3$ contributions in Sect. 4.2. Accordingly, increases of the $NO_x$ emissions from land transport lead to an increase of $OH^{TRA}$ and vice versa. Besides the changes of $NO_x$, also changes of the VOC or CO emissions will affect OH. Contributions of land transport emissions to CO and VOC are, however, not analysed in detail in the present study, because it is beyond the scope of the present study.

As for $O_3^{SHP}$, $OH^{SHP}$ is mainly limited to the lower troposphere (see Fig. 14). The maxima are simulated near the surface, with enhanced contributions up to 500 hPa. In the tropical upper troposphere only small contributions are simulated, because also $O_3^{SHP}$ is limited to the lower troposphere (see Sect. S3 in the Supplement). In 2015, the zonal average of $OH^{SHP}$ reaches up to 16 fmol mol$^{-1}$ near the surface and decreases to $8-12$ fmol mol$^{-1}$ in the middle and upper troposphere.

In accordance with the changes of the emissions in the three SSP projections, the contributions of $OH^{SHP}$ is lowest in SSP1-1.9, followed by SSP3-7.0 and SSP2-4.5, which has the largest contributions in 2050. The contributions in 2050 with SSP2-4.5 emissions, however, are lower compared to 2015.

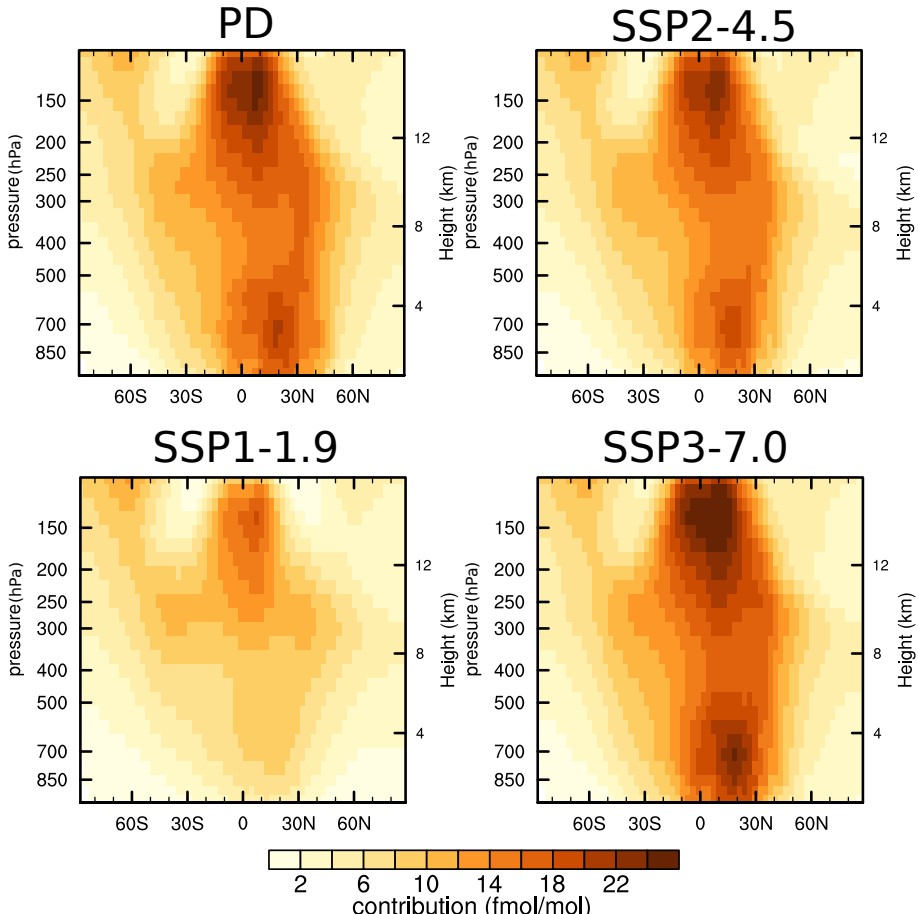

**Figure 13.** Zonal average of the absolute contribution of land transport emissions to OH (in $\text{fmol mol}^{-1}$ for PD, SSP1-1.9, SSP2-4.5 and SSP3-7.0. The shown values are 5 year averages.

The zonal average of $\text{OH}^{\text{AIR}}$ (see Fig. 15) shows a clear peak in the Northern hemisphere between $30-60°\text{N}$ and $200-300\,\text{hPa}$ which is in accordance with Rieger et al. (2018). The region with the maxima of $\text{OH}^{\text{air}}$ is similar with the region of the largest impact of aviation emissions on OH at around $40°\text{ N}$ and between $200-300\,\text{hPa}$ reported by Hoor et al. (2009). In 2015 the zonal average of $\text{OH}^{\text{air}}$ peaks up to $22\,\text{fmol mol}^{-1}$. In agreement with the changes of the aviation emissions and $\text{O}_3^{\text{AIR}}$, $\text{OH}^{\text{air}}$ increases in 2050 for SSP2-4.5 (up to $36\,\text{fmol mol}^{-1}$) and SSP3-7.0 (up to $33\,\text{fmol mol}^{-1}$) compared to 2015. Due to the large 610 decrease of emissions in SSP1-1.9, peak values of the zonal average of $\text{OH}^{\text{AIR}}$ decrease to $13\,\text{fmol mol}^{-1}$.

    As OH is strongly coupled to $\text{O}_3$ (and vice-versa) also OH shows a similar non-linearity as $\text{O}_3$. Accordingly, also the OH contributions respond non-linearly to the changes of the emissions from the traffic sectors (see Sect. S7 in the Supplement).

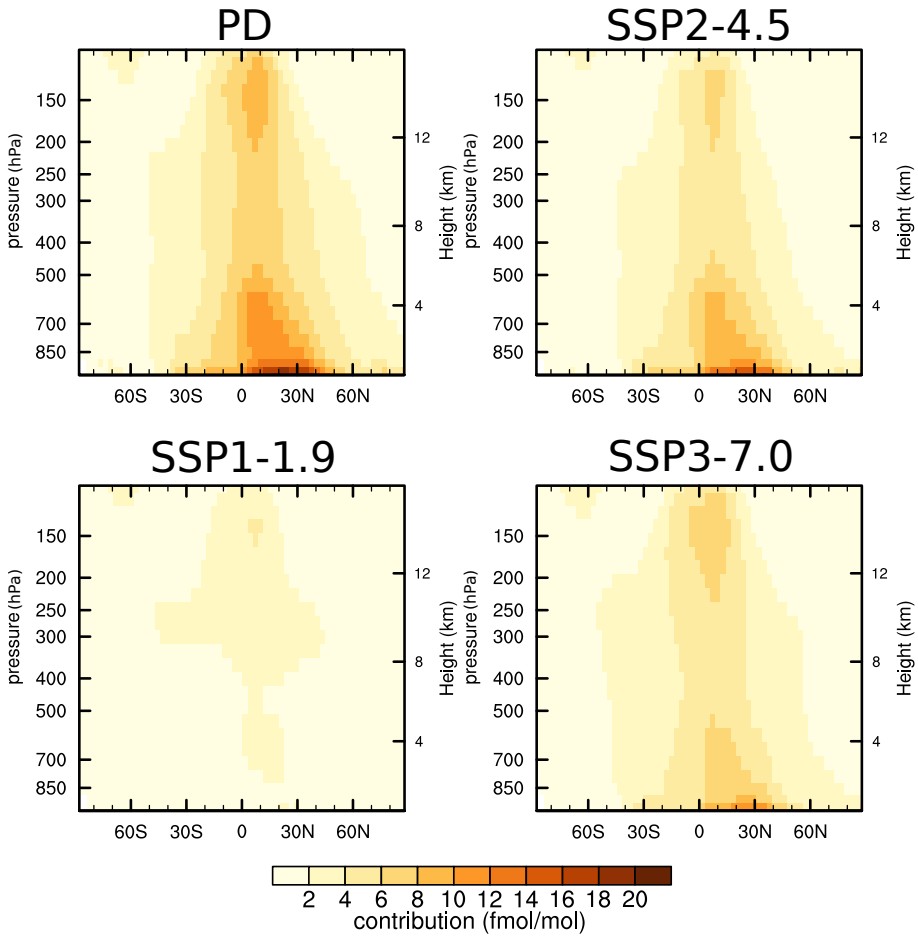

**Figure 14.** Zonal average of the absolute contribution of shipping emissions to OH (in $\text{fmol mol}^{-1}$ for PD, SSP1-1.9, SSP2-4.5 and SSP3-7.0. The shown values are 5 year averages.

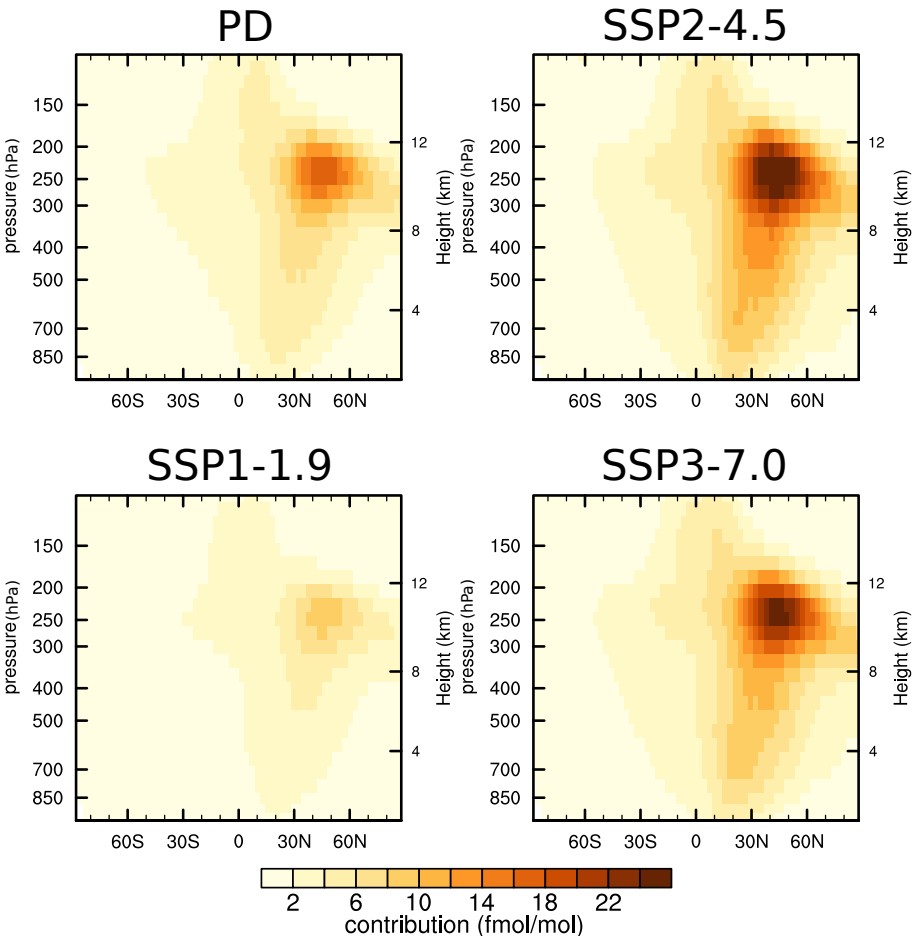

**Figure 15.** Zonal average of the absolute contribution of aviation emissions to OH (in fmol mol$^{-1}$ for *PD*, *SSP1-1.9*, *SSP2-4.5* and *SSP3-7.0*. The shown values are 5 year averages.

## 6.2 Reductions of the methane lifetime

Methane is an important greenhouse gas and therefore changes in its lifetime, e.g. due to changes in OH caused by emissions of the transport sector, might contribute significantly to the climate effect of transport emissions. Based on the contributions of the different emissions sectors to OH, the effect of the different emission source onto the methane lifetime is analysed. This analyses helps us to understand how the transport emissions affect the methane lifetime. In the following we define methane lifetime as the yearly global mean of $CH_4$ divided by the yearly global mean (expressed as overbars) of the loss of $CH_4$ by OH as diagnosed by TAGGING ($loss(CH_4)_{total}$, see Rieger et al., 2018);

$$\tau(CH_4) = \frac{\overline{CH_4}}{loss(CH_4)_{total}} \tag{6}$$

The lifetime is calculated for the troposphere, i.e. below the monthly mean tropopause (pressure) height. In the present study the tropopause is defined according to the WMO definition based on the temperature lapse rate for latitudes equatorward of $30°$, and as the potential vorticity iso-surface of 3.5 PVU at latitudes poleward of $30°$ (see Grewe and Dameris, 1996, for an analysis on the effects of different tropopause definitions).

The loss rates of $CH_4$ for 18 different sectors, including the transportation sectors land, ship, and aviation are diagnosed based on the OH contribution discussed in Sect. 6.1. Based on these OH contributions, the relative reduction of the methane lifetime of a specific emission source (for example shipping emissions) is then calculated as:

$$\Delta_{rel}\tau(CH_4)_{SHP} = \frac{\overline{\tau(CH_4)_{total}} - \overline{\tau(CH_4)_{no-SHP}}}{\overline{\tau(CH_4)_{no-SHP}}} = -\frac{\overline{loss(CH_4)_{SHP}}}{\overline{loss(CH_4)_{total}}}. \tag{7}$$

'SHP' is the $CH_4$ loss attributed to the emissions of the shipping sector based on the $OH^{SHP}$, 'no-SHP' is the loss of all other categories, except the shipping sector. Please note, the reciprocals of the methane lifetime calculated for each tagged OH are additive and sum up to the total methane lifetime as calculated by Equation 6. The relative reduction as calculated according to Equation 7 always refer to the total methane loss for one specific simulation. Thus, the relative reduction of the methane lifetime for one simulation sum up to 100 %. In the different simulations, the methane lifetime differs slightly. This needs to be considered when comparing the relative reductions of the methane lifetime analysed for the different simulations. The methane lifetimes analysed for the different simulations by Equation 6 are 7.46 a (2015), 7.45 a (SSP2-4.5 in 2050), 7.43 a (SSP3-7.0 in 2050) and 8.01 a (SSP1-1.9 in 2050).

Figure 16a shows the relative reduction of the methane lifetime (calculated with Equation 7) due to the emissions of the different transport sectors for 2015 and the three emissions projections in 2050. Generally, transport emissions lead to a reduction of the methane lifetime and hence to a reduction of the climate effects of methane. From all transport modes. the land transport emissions have the largest effect on the methane lifetime. The relative reductions of the methane lifetime by land transport emissions ($\Delta_{rel}\tau(CH_4)_{TRA}$) ranges between -7 % (SSP1-1.9) and -16 % (SSP3-7.0). The relative reductions of the methane lifetime from shipping emissions ($\Delta_{rel}\tau(CH_4)_{SHP}$) range from -9 % (*PD*) to -2 % (SSP1-1.9). The relative reduction of the

methane lifetime of aviation emissions ($\Delta_{rel}\tau(CH_4)_{AIR}$) is lowest for SSP1-1.9 (-3 %) and largest for SSP2-4.5 (-7 %) and SSP3-7.0 (-6 %) emissions. Despite for 2015 conditions, shipping and aviation have similar relative reductions of the methane lifetime, even though the shipping sector has much larger emissions (see Fig. 1). This is due to the larger mixing ratios of $OH^{AIR}$ compared to $OH^{SHP}$ (see Sect. 6.1). Moreover, due to the strong non-linearity of the OH chemistry, even in the clean SSP1-1.9 projection the transport sectors, especially the land transport sector, have a rather large effect on the methane lifetime. This non-linearity is larger for the reduction of the methane lifetime than for the ozone RF. As example, the ozone RF of land transport decreased by around 70 % between 2015 and SSP1-1.9, while the relative reduction of the methane lifetime is 50 %. The reverse effect can be observed when increasing transport emissions, for example the ozone RF of land transport emissions between 2015 and SSP3-7.0 increase by 15 %, while the reduction of the methane lifetime changes by 8 %.

Figure 16b breaks down the reduction of the methane lifetime by land transport emissions to the different geographical regions. In agreement with the contributions to the ozone RF (see Sect. 5) the largest relative reduction of the methane lifetime are caused by land transport emissions from the rest of the world (LT-ROW). Relative reductions of the other regions are smaller by a factor of 2−3, with the largest relative reductions from South Asia (LT-SA).

Generally, the relative reductions of the methane lifetime as analysed by the source attribution approach are much larger than relative reductions of the methane lifetime calculated by a 5 % perturbation (see Sect. S5 in the Supplement for a detailed analysis for 2015 conditions). The reduction of the methane lifetime as analysed by the perturbation approach for the land transport emissions is -1.9 %, -4.5 %, for the shipping emissions and -1.1 % for the aviation emissions, respectively. Accordingly, the reductions of the methane lifetime diagnosed by the source attribution approach are larger by a factor of 7.7 (land transport), 1.9 (shipping) and 3.5 (aviation) compared to the reductions as analysed by a 5 % reduction with the perturbation approach. This large difference of the relative reductions of the methane lifetime as analysed by the two methods also leads to shifts in the ranking of the different transport sectors w.r.t. the influence of the methane lifetime. While with the perturbation method the shipping emissions are a factor two larger than for land transport emissions, with the source attribution method the contribution of land transport emissions is a factor 2 larger than for shipping emissions. The main reason for this different behaviour is that most shipping emissions take place in cleaner environments with less emissions compared to the land transport emissions. In these cleaner environments the non-linearity of the ozone chemistry is not that pronounced and the linearisation of the perturbation approach does not affect the results strongly (see also Grewe et al., 2010).

## 7 Limitations and uncertainties

In the following we discuss the limitations and uncertainties of this study. The main points we discuss are the usage of the same meteorological conditions and methane emissions in all simulations, the different available tagging method for source attribution and the influence of the errors in the geographical distribution of the aviation emissions in CMIP6.

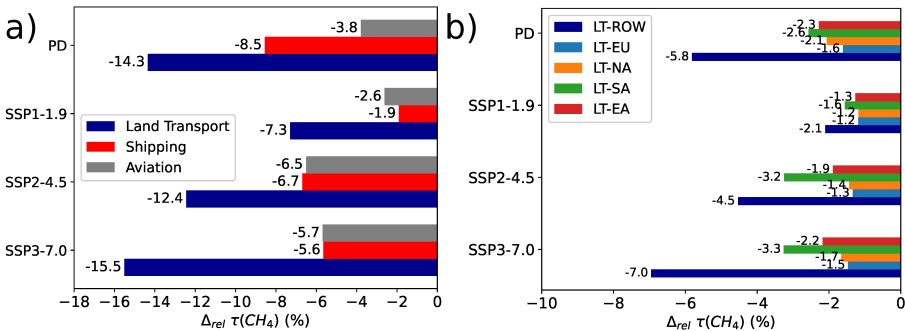

**Figure 16.** Methane lifetime reduction by different emission sectors based on their contributions to OH. a) Sectors land, aviation, ship, and for different SSPs, b) total transportation (all transport sectors) from different regions and for different SSPs. Note: Lifetime reductions due to individual sectors are additive. See the text for a description of the calculation. All values are 5 year averages.

## 7.1 Effects of climate change

The signals of emissions from specific regions or specific emission sources (e.g. from aviation) are small. To quantify these signals, we apply the QCTM mode (see Sect. 2) in which chemistry and dynamics are decoupled. Accordingly, the dynamics (and therefore the climatic state) is identical in every model simulation. This approach of applying present day dynamics for future emission scenarios is commonly used when the effects of certain emission changes or sources on the atmospheric chemistry are investigated on the global and regional scale (e.g., Eyring et al., 2007; Hoor et al., 2009; Hodnebrog et al., 2012; Righi et al., 2015; Matthias et al., 2016).

Due to this approach, however, our model simulations do not consider changes of meteorology and climate between 2015 and 2050. Accordingly, emissions which are based on meteorological conditions (e.g. biogenic emissions, lightning-$NO_x$) are identical in all simulations. With climate change, these emissions are likely to increase (von Schneidemesser et al., 2015). This increase could alter the contributions of the anthropogenic emissions, for instance increased biogenic VOC emissions may affect the ozone production efficiency, while increased lightning-$NO_x$ in the upper troposphere may compete with $NO_x$ emissions from the aviation sector.

Moreover, increased biogenic emissions and changed atmospheric conditions (e.g. increased temperature and it's effects on kinetics) likely lead to an increase of ozone near highly polluted regions (knows as 'climate-penalty', Zanis et al., 2022). In addition, climate change likely leads to an decrease of ozone in remote regions due to the increase of water vapour (known as 'climate-benefit', Zanis et al., 2022). In addition, during periods of droughts and heat-waves, reduced ozone deposition to vegetation could increase ground-level ozone (Lin et al., 2020).

Altogether, this could affect also the contributions of the traffic emissions. A reduced life-time of ozone, especially over the oceans, would likely lead to a reduction of ozone attributable to shipping emissions. Also long-range transport, especially the source-receptor relationships, might be affected by changes of the ozone lifetime. At the same time, the increase of ozone in polluted regions in a changing climate could affect ozone contributions especially from land transport emissions.

Koffi et al. (2010) considered the effects of climate change on the ozone effects of transport emissions applying a 5 % emissions reduction (i.e. with the perturbation approach). Globally, they report a small decrease of the ozone changes caused by transport emissions due to climate change, but with strongly varying regional patterns. The effect of climate change on ozone contributions (i.e. applying a tagging approach) needs to be analysed in follow-up studies.

## 7.2 Land-use change

Similar to changes of the climate, changes in land-use also affect ozone (e.g. Wu et al., 2012; Wang et al., 2020). The effects of land-use change on ozone are due to various processes, such as changes of biogenic emissions, effects on dry deposition, and changes of temperature (e.g., by effects on radiation and evapo-transpiration). Land-use change, and the corresponding effects, also heavily depend on the considered scenario (Popp et al., 2017). These effects are not considered to isloate the effects of the emissions changes only.

## 7.3 Fixed lower boundary condition for methane

To be able to estimate only the effect of the changing anthropogenic emissions of short lived species, we also kept the lower boundary conditions for methane fixed in all model simulations. We chose this approach to be able to attribute and understand the effects of the changes of the other ozone precursors in more detail. Increasing methane levels strongly affect atmospheric chemistry. In addition, increasing methane levels can have very complex effects on ozone produced form a specific emission
sources. As an example, Butler et al. (2020) reported a strong increase of ozone attributable to shipping emissions with increasing methane levels, while the contribution of other emission sources showed a smaller increase. To suppress such complex effects, we kept the $CH_4$ lower boundary condition constant in all simulations. To investigate how increased methane levels would affect the results, we performed an additional sensitivity study for 2050 taking short-lived emissions and lower boundary conditions for methane from the SSP2-4.5 projection for 2050. In this simulation, globally averaged prescribed methane
mixing ratios at the surface are around 9 % larger compared to the *SSP2-4.5* simulation. This increase leads to $\approx 2.5$ % (or 0.9 DU) increased globally averaged tropospheric ozone columns. Similar as reported by Butler et al. (2020), the contributions show a very heterogeneous response. The contribution to tropospheric ozone of emission categories with strong $NO_x$ emissions increases, with the largest increases by the categories lightning (3.7 %), $N_2O$ degradation (3.1 %), aviation (2.9 %), and shipping (2.4 %). In addition, ozone attributed to methane degradation also increases by 9.1 % (see also Sect. S6 in the
Supplement). Moreover, the response of the methane increase in our study differs from the results reported by Butler et al. (2020), which is caused by differences in the tagging approach (see next paragraph) and by different emissions inventories and different magnitude of the methane increases. However, the analysis shows the importance of also considering changes of the methane lower boundary condition/emissions. Especially when considering our results of the SSP3-7.0 this effect should be kept in mind, because SSP3-7.0 shows even larger methane levels compared to SSP2-4.5.

## 7.4 Source apportionment method

The tagging method which we applied for source attribution is a diagnostic method. The method itself is mathematically correct, but it has some simplifications which influence the results, as discussed in detail by Mertens et al. (2018) and Butler et al. (2018). In addition, different ozone source attribution methods exist (e.g. Lelieveld and Dentener, 2000; Emmons et al., 2010; Butler et al., 2018). Our approach attributes $NO_x$ and VOC emissions at the same time, while the approach of Butler et al. (2018) either tags $NO_x$ or VOC. Especially for the shipping sector, but most likely also for the aviation and the land transport sector, which all have rather large $NO_x$ emissions and low VOC emissions, our approach leads to lower contributions compared to the $NO_x$ tagging approach (see also Butler et al., 2020).

## 7.5 Discrepancy between geographical distributions of the aviation emissions

We used the original aviation emissions from CMIP6. Thor et al. (2023) reported a discrepancy in the geographical distribution of the aviation emissions in CMIP6. This discrepancy leads to overestimated emissions in Arctic and Antarctic regions and underestimated emissions in the tropics. Correcting for this discrepancy in the 2015 emissions leads to an around 8% larger ozone RF and 6 % larger relative reduction of the methane lifetime of the aviation emissions compared to the values discussed in the present manuscript (see Thor et al., 2023, for detailed analyses).

## 7.6 Future approaches

Future studies should aim to consider the effects of changed methane emissions/lower boundary conditions and the effects caused by climate and land-use change on the ozone and OH contributions of transport emissions and the associated radiative forcings and reductions of the methane lifetime. Due to the effects discussed above, such studies need a much more complicated design of the numerical experiments. Moreover, much longer integration times needs to be taken into account to yield signals of the transport emissions larger than the internal variability (see for example **?**). As the response of atmospheric chemistry, and more importantly also natural emissions, of climate change vary (e.g. Turnock et al., 2020) a multi-model approach would be of interest here, however only few models feature ozone tagging approaches as used in the present study.

## 8 Conclusions and Outlook

This manuscript presents an analysis of the effects of transport emissions from the sectors land transport, aviation, and shipping on ozone and methane lifetime, because the transport sector is an important source for ozone precursors. The aim is to support efforts to reduce transport emissions with a better understanding of the highly non-linear response of atmospheric chemistry on these emission changes. We investigated present day conditions for 2015 and three projections of the emissions for 2050 according to the Shared Socioeconomic Pathway (SSPs) SSP1-1.9, SSP2-4.5 and SSP3-7.0. To do so, we performed a set of 5 year long simulations with the chemistry-climate model EMAC. Atmospheric dynamics are kept identical in order to quantify the small signals of specific transport sectors. For the source attribution of the emission effects on ozone, OH, $HO_2$ and ozone

related precursors, we used a tagging method. Based on the contributions of the transport emissions to ozone and OH, we are able to calculate the radiative forcings of ozone attributable to these emissions. We also introduce a novel calculation of the relative reduction of the methane lifetime for all transport sectors based on the OH contributions whereas previous studies used the so-called perturbation approach.

Our analyses of the model results can be summarised in the following most important findings:

- **Changes of transport emissions in SSP**: Globally, transport emissions are reduced strongly in the SSP1 projection, following the high share of public transport, the strong technological advancements including adaption of newer technologies, increased use of high speed trains and overall reduced demand for transportation assumed in this scenario. In the SSP2 projection the emissions of the land transport sector decrease compared to 2015 in most regions except South Asia. Here, the large increase in demand for mobility caused by increasing population and insufficient pollution control and limited technological advancements lead to an increase of land transport emissions. Also global $NO_x$ emissions from aviation are doubled compared to 2015 due to increasing demands, while the shipping emissions are reduced slightly. In the SSP3 projection only Europe and North America show a larger decrease of land transport emissions in 2050 compared to 2015, while emissions increase in most other regions. The aviation emissions are similar as in the SSP2 projection and shipping emissions are reduced compared to SSP2 and to 2015, most likely due to international trade barriers in the narrative of the scenario.

- **Efficiency of ozone formation:** Due to the non-linearity of atmospheric chemistry, the response of atmospheric composition to the changed emissions is rather complex. In accordance with previous publications, we find a large efficiency of the ozone formation from aviation emissions, while ozone from land transport emissions in Europe and North America is formed rather inefficient. A similar behaviour is also observed for OH, which is strongly bound to the $NO_x$-$O_3$ chemistry.

- **Atmospheric effects of SSP1 emissions:** Due to the strong decrease of emissions in SSP1, the non-linearity of the $NO_x$-$O_3$ chemistry compensates for some of the emission reductions. However, the total decrease of the emissions is so large that the contributions of the transport emissions to ozone and the associated radiative forcing decrease strongly. The non-linearity of the OH-response and the associated relative reduction of the methane lifetimes is stronger as for the $O_3$ response. Accordingly, the (cooling) effect of the transport emissions on the climate by the decrease of the methane lifetime can get more important compared to the warming effect of $O_3$, because the reduction of the methane lifetime is larger as the increase of ozone.

- **Atmospheric effects of SSP2 emissions:** In the SSP2 projection the land transport emissions are reduced strongly in many regions, leading to a large reduction of the ozone contributions and associated radiative forcings. However, ozone formed from land transport emissions over South Asia have rather large radiative forcings as this ozone is transported in the upper troposphere very efficiently. Therefore, the radiative forcing from global land transport emissions is reduced by only 15 %. Although the total $NO_x$ emissions of the aviation sector are low compared to the land transport sector, the

doubling of the aviation $NO_x$ emissions in SSP2 compared to 2015 almost compensates for the reductions in the land transport and the shipping sector in terms of ozone RF. Accordingly, the total radiative forcing of the transport sector is similar in 2050 for SSP2 compared to 2015. The contributions to the methane life-time show a similar behaviour.

– **Atmospheric effects of SSP3 emissions:** In the SSP3 projection land transport emissions in Europe and North America are reduced, but the non-linearity of the ozone chemistry strongly compensates for this reduction and the ozone contributions are not reduced likewise. In addition, due to the lack of international co-operations and increasing emission in other parts of the world, the contribution of ozone from land transport transported via long-range transport increases over Europe and North America with SSP3 in 2050 compared to 2015. The global increase of land transport emissions, especially in South Asia, leads to an increase of the ozone radiative forcing from land transport emissions. Moreover, also the doubling of aviation emissions compared to 2015 leads to an overall larger ozone radiative forcing and larger contributions on the methane lifetime of the transport sector in 2050 compared to 2015. Especially for the results of SSP3 it should kept in mind that we apply present day methane-levels in all simulations. Applying the methane levels for SSP3 in 2050 likely leads to even larger ozone increases, but the responses of the different emission sectors on the methane increase are very complex and require further investigations in follow up studies.

– **Contributions of the transport sectors on** $O_3$ **and** OH**:** Overall, the land transport sector is the transport sector with the largest emissions and the largest effects on ozone and OH. In 2015, this is followed by shipping and aviation. Due to the increase of aviation emissions in SSP2 and SSP3 compared to 2015, caused by increased demand, and the reduction of shipping emissions in SSP2 and SSP3 compared to 2015, the effects of the aviation sector to ozone radiative forcing and methane lifetime are larger compared to the shipping sector.

– **Contributions from South Asia:** Already for present day, land transport emissions from South Asia strongly contribute to tropospheric ozone. This contribution is projected to increase in SSP2 and SSP3. Given the large radiative efficiency of ozone formed from this emissions and their negative effects on air quality, very urgent mitigation policies aiming at a reduction of land transport emissions in this region are needed.

– **Implications for mitigation option:** The complex, non-linear atmospheric responses of $O_3$ and OH highlight the necessity to investigate effects of possible mitigation options for the transport sector using state of the art atmospheric chemistry models. These effects are overlooked if only emission reductions are analysed.

– **Tagging vs. perturbation approach** The application of the source attribution method for OH and the diagnosed reductions of the methane lifetime yields to results which deviate strongly from the results of the perturbation approach. With the 5 % perturbation the shipping sector leads to a larger reduction of the methane lifetime than the land transport and aviation sector. With the source attribution method emissions from the land transport sector lead to the largest reduction of the methane lifetime (around 7 times larger compared to a 5 % perturbation), followed by the shipping and the aviation sector. Such larger reductions of the methane lifetime also influence the total radiative forcing caused by the $NO_x$ emissions which will be analysed in a follow up study.

Due to the very complex response of atmospheric chemistry to emission changes, we did not consider the effects of climate change in our simulations. Also the effects of increased methane emissions was investigated in a short sensitivity simulation only. Future studies should incorporate both effects. Especially, for example changes in lightning-$NO_x$ emissions and biogenic emissions could also strongly influence contributions of land transport emissions to the atmospheric composition. Reduction of the ozone lifetime due to increased water vapour could reduce long-range transport or contributions from shipping. Moreover, in very strong mitigation cases such as the SSP1, the role of these natural emissions will be more and more important due to the non-linearity of the ozone chemistry, as the effects of the transport emissions strongly depend on the background conditions. Such a study, however, would need much longer model integration times and due to different responses of atmospheric composition by climate change by different models a multi-model approach would be very valuable here. The lack of publicly available information on the detailed assumptions and trends in the transport sectors from the SSP narratives complicated the detailed interpretation of some of the results. For future studies it would therefore be very valuable if such important information on the developments in the transport sector would also be available publicly (e.g. for CMIP7).

*Code and data availability.* MESSy is continuously developed and applied by a consortium of institutions. MESSy and the source code are licensed to all affiliates of institutions which are members of the MESSy Consortium. Institutions can become members of the MESSy Consortium by signing the MESSy Memorandum of Understanding. More information can be found on the MESSy Consortium website (http://www.messy-interface.org, last access: 27 November 2023). The model configuration discussed in this paper is based on EMAC version 2.55. The exact set-up used to produce the results of this paper is archived at the German Climate Computing Center (DKRZ) and can be made available to members of the MESSy community upon request. The model results analysed in this study are available on request and will be made available during the peer-review process.

*Author contributions.* M.M. performed the model simulations and analysed the model results , S.B. assisted with the emissions processing and performed the methane lifetime analyses, V.G. and V.R. developed the improved $HO_x$ TAGGING needed for this study, J.H. designed and coordinated the experimental setup, P.J. supported the necessary model development and methane lifetime analyses, P.G. supported the model setup and emission processing, A.L. performed analyses of the ozone non-linearity and emission, S.M., V.G. and R.N.T. helped to analyse the aircraft emissions and M.R. supported the discussion of the experimental setup and the analysis of the traffic projections in the SSPs. All authors supported the interpretation of the modelling results and contributed to the writing of the manuscript.

*Competing interests.* The authors declare that no competing interests are present.

*Acknowledgements.* We thank Helmut Ziereis (DLR) and Anja Schmidt (DLR) for very helpful discussions and comments to the manuscript. Moreover we thank Moritz Menken (DLR) for support with the the SWOOSH data. We thanks Owen Cooper, Michael Prather and two anony-

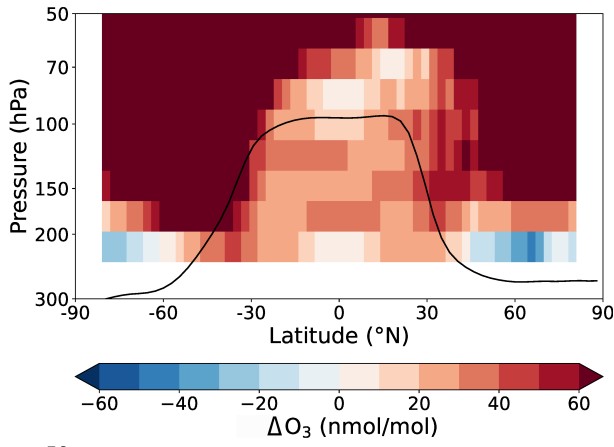

**Figure A1.** Absolute difference (in $\mathrm{nmol\,mol^{-1}}$) between ozone mixing ratios from the PD simulation and the SWOOSH data. The SWOOSH data are interpolated horizontally onto the coarser EMAC grid. The EMAC data are interpolated vertically onto the coarser SWOOSH grid. All data are averages for $2013-2017$. The colour scale is optimized for tropospheric ozone values and not for stratospheric values.

mous referees for their very helpful comments which improved the manuscript considerably. This work uses resources of the Deutsches Klimarechenzentrum (DKRZ) granted by its Scientific Steering Committee (WLA) under project id 80. Further, datasets provided by MESSy via the DKRZ data pool were used. his study was supported by the DLR transport research programme (TraK and DATAMOST projects). Moreover, this work has received funding from the European Union's Horizon 2020 research and innovation programme under grant agreement No 875036 (ACACIA) and from the German Federal Ministry of Education and Research (Funding Nr.: 01LN2207A, IMPAC$^2$T).

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
