# Peer review of "The contribution of transport emissions to ozone mixing ratios and methane lifetime in 2015 and 2050 in the Shared Socioeconomic Pathways (SSPs)"

_EGUsphere, 2024_

## Referee Comment (RC3)

**Reviewer Comments on Manuscript:**

"The Contribution of Transport Emissions to Ozone Mixing Ratios and Methane Lifetime in 2015 and 2050 in the Shared Socioeconomic Pathways (SSPs)"

**General comments:**

The manuscript offers a comprehensive assessment of transport emissions' impacts on ozone and the hydroxyl radical (OH) across different transport sectors using the EMAC Chemistry-Climate model's simulations. The study uses an innovative approach to quantify contributions to OH for the years 2015 and projections for 2050, under various SSPs. The analysis extends to ozone radiative forcing and methane lifetime reductions, highlighting the manuscript's value to environmental policy and planning stakeholders.
The manuscript is well written and provides an extensive analysis of the impacts arising from various emission scenarios. However, there are several areas where further development could enhance the study's robustness. After addressing the suggestions outlined below, the manuscript should be considered for publication due to its valuable contribution to the field.

**Model Evaluation:**

The manuscript would greatly benefit from a dedicated validation section. Such a section should detail the model's proficiency in simulating the chemical environment and meteorology for the base year of 2015. This could include comparisons of model outputs with observed data or results from prior studies to establish the model's skills.

**Model Description:**

The description of the EMAC model setup, including its chemical mechanisms, is thorough. Nonetheless, the manuscript would benefit from additional details on the model's parameterizations, particularly those influencing ozone and other chemical species. This should encompass radiation, deposition, and boundary layer schemes, as well as the land surface model used. A clarification on whether the simulations incorporate direct radiation feedback would be pertinent.

**Introduction and Methods:**

**Structural Suggestions:**

Consider relocating parts of the limitations and uncertainties discussion, currently in line 610, to the introduction or methodology sections. This would help set the reader's expectations early in the manuscript.

**Analysis and Discussion:**

**Climate Change Impact:**

Incorporating a discussion on the potential changes and impacts of climate change on atmospheric chemistry and transport patterns is recommended. This includes a thorough consideration of radiation feedbacks and their prospective effects on future climate change scenarios. For example, changes in surface and atmospheric temperatures can profoundly influence ozone chemistry; as temperatures increase, so do evaporation rates, which lead to a higher concentration of water vapor in the atmosphere, potentially affecting OH and ozone levels. Additionally, changes in cloud cover can alter photolysis rates, thereby impacting ozone formation and destruction.

**Land Surface Model Considerations:**

It is important to discuss the implications of land surface model choices within the simulations. Soil moisture variability, alterations in land use, and vegetation cover driven by climate scenarios play an important role in the soil's chemical processes and the land's overall energy budget. For instance, soil moisture dependent on the chosen climate scenario affects soil chemistry, influencing how land surface models simulate these processes. Similarly, changes in land use and vegetation cover have the potential to modify the absorption, reflection, and emission of radiant energy at the land surface. Moreover, the type of vegetation and temperature changes can affect the deposition of chemical species and their uptake by plants.

**Sensitivity Analysis:**

Could a sensitivity analysis be performed to evaluate how various model assumptions, such as chemical reaction rates and deposition processes in a changing climate, might affect the outcomes? This would contribute to understanding the study's conclusions' robustness.

---

## Community Comment (CC2)

Comments by Owen R. Cooper (TOAR Scientific Coordinator of the Community Special Issue) on:

**The contribution of transport emissions to ozone mixing ratios and methane lifetime in 2015 and 2050 in the Shared Socioeconomic Pathways (SSPs)**

Mariano Mertens, Sabine Brinkop, Phoebe Graf, Volker Grewe, Johannes Hendricks, Patrick Jöckel, Anna Lanteri, Sigrun Matthes, Vanessa S. Rieger, Mattia Righi, and Robin N. Thor

EGUsphere [preprint], https://doi.org/10.5194/egusphere-2024-324, 2024.
Discussion started: 12 Feb 2024;  Discussion closes April 20, 2024

This review is by Owen Cooper, TOAR Scientific Coordinator of the TOAR-II Community Special Issue. I, or a member of the TOAR-II Steering Committee, will post comments on all papers submitted to the TOAR-II Community Special Issue, which is an inter-journal special issue accommodating submissions to six Copernicus journals:  ACP (lead journal), AMT, GMD, ESSD, ASCMO and BG. The primary purpose of these reviews is to identify any discrepancies across the TOAR-II submissions, and to allow the author teams time to address the discrepancies.  Additional comments may be included with the reviews. While O. Cooper and members of the TOAR Steering Committee may post open comments on papers submitted to the TOAR-II Community Special Issue, they are not involved with the decision to accept or reject a paper for publication, which is entirely handled by the journal's editorial team.

**General Comments:**

TOAR-II has produced two guidance documents to help authors develop their manuscripts so that results can be consistently compared across the wide range of studies that will be written for the TOAR-II Community Special Issue.  Both guidance documents can be found on the TOAR-II webpage: https://igacproject.org/activities/TOAR/TOAR-II

*The TOAR-II Community Special Issue Guidelines*:   In the spirit of collaboration and to allow TOAR-II findings to be directly comparable across publications, the TOAR-II Steering Committee has issued this set of guidelines regarding style, units, plotting scales, regional and tropospheric column comparisons, tropopause definitions and best statistical practices.

*Guidance note on best statistical for TOAR analyses*:  The aim of this guidance note is to provide recommendations on best statistical practices and to ensure consistent communication of statistical analysis and associated uncertainty across TOAR publications. The scope includes approaches for reporting trends, a discussion of strengths and weaknesses of commonly used techniques, and calibrated language for the communication of uncertainty. Table 3 of the TOAR-II statistical guidelines provides calibrated language for describing trends and uncertainty, similar to the approach of IPCC, which allows trends to be discussed without having to use the problematic expression, "statistically significant".

**Major Comments:**

This paper provides a thorough analysis of the impact of transport emissions on present-day and future (2050) ozone based on three different SSPs.  This is a complex endeavor requiring a wide range of tagged tracer runs and sensitivity tests, and it's not possible to consider every situation and account for every competing process (e.g. emissions, climate change, non-linearity).  The authors are of course aware of this challenge and provide some extensive discussion in Section 7.  I think this section would benefit from some further discussion regarding SSP3-7.0 and the expected impacts of climate change and increasing methane concentrations, as assessed in Chapter 6 of IPCC AR6 WG-I (Szopa et al., 2021).

Figure 6.4 in Szopa et al. shows an increase of the tropospheric ozone burden of roughly 10% from 2014 to 2050, based on SSP3-7.0, and much of this increase is due to projected increases in methane. Figure 6.20 in Szopa et al. indicated average ozone increases across South Asia of 8-10 ppb by 2050, under SSP3-7.0. These ozone increases seem to be much larger than your projected increases, as shown in your Figure 2. Part of this discrepancy could be due to differences in methane concentrations, as you discussed in Section 7. But another likely explanation is the ozone climate penalty that impacts boundary layer ozone, as discussed by Zanis et al. 2022. Your paper does not mention the climate penalty and I think that it deserves some discussion. Another important finding of IPCC AR6 and Zanis et al. (2022) is that a warmer climate will be more humid, especially in the boundary layer, which will lead to a reduction of ozone lifetime in remote regions, such as over the oceans. Your Figure 2 does not show a consistent reduction of ozone across the oceans under SSP3-7.0, probably because you use the same meteorology in 2015 and 2050; some discussion of this phenomenon would also be helpful.

Figure 5. Given that SSP1-1.9 has strongly decreasing transport emissions in all regions, I am surprised that none of the regional reductions produces ozone reductions in downwind regions. Why are there no ozone reductions in the receptor regions?

Figure 7. If the future scenarios included climate change, with more humidity in the boundary layer and therefore a shorter ozone lifetime, would the ozone reductions due to shipping emissions reductions be even more pronounced?

Section 4.4
A recent paper by Wang et al. (2022) indicates that the impact of aviation on the global tropospheric ozone burden is greater than suggested by previous studies. How does your analysis compare to that of Wang et al.?

**Minor Comments:**

Figure S4. There is hardly any difference in surface ozone between PD and SSP3-7, which is surprising. SSP3-7 is projected to have an increase in the tropospheric ozone burden, especially in the free troposphere. This should mean that ozone at high elevations sites (Greenland, the western USA, Tibetan Plateau, the Andes, Antarctica) should be higher under SSP3-7, but they appear to be almost the same. Is this due to your 2015 and 2050 simulations having the same methane concentrations, instead of higher methane in 2050?

Line 622
When considering the impact of climate change on ozone, a relevant study is Lin et al. 2020, who show that drought and heat waves can limit ozone deposition to vegetation.

Line 410
When discussing ozone non-linearity, two relevant studies are Wu et al. (2009) and Wild et al. (2012). Similarly, when discussing differences in ozone production efficiency among regions, the study by Zhang et al. (2016) is very important as it demonstrated that ozone production efficiency is much greater in tropical regions than at northern mid-latitudes.

**References:**

Lin, M., Horowitz, L.W., Xie, Y., Paulot, F., Malyshev, S., Shevliakova, E., Finco, A., Gerosa, G., Kubistin, D. and Pilegaard, K., 2020. Vegetation feedbacks during drought exacerbate ozone air pollution extremes in Europe. Nature Climate Change, 10(5), pp.444-451.

Wang, H., et al. (2022), Global tropospheric ozone trends, attributions, and radiative impacts in 1995–2017: an integrated analysis using aircraft (IAGOS) observations, ozonesonde, and multi-decadal chemical model simulations, Atmos. Chem. Phys., 22, 13753–13782, https://doi.org/10.5194/acp-22-13753-2022

Wild, O., et al.: Modelling future changes in surface ozone: a parameterized approach, Atmos. Chem. Phys., 12, 2037–2054, https://doi.org/10.5194/acp-12-2037-2012, 2012.

Wu, S., Duncan, B.N., Jacob, D.J., Fiore, A.M. and Wild, O., 2009. Chemical nonlinearities in relating intercontinental ozone pollution to anthropogenic emissions. Geophysical Research Letters, 36(5).

Zanis et al., 2022, Climate change penalty and benefit on surface ozone: a global perspective based on CMIP6 Earth System Models, Environmental Research Letters, 17

Zhang, Y., et al. (2016), Tropospheric ozone change from 1980 to 2010 dominated by equatorward redistribution of emissions, *Nature Geoscience*, 9(12), p.875, doi: 10.1038/NGEO2827

---

## Author Comment (AC1)

Dear referee#1,
we thank you very much for your in-depth review of our manuscript egusphere-2024-324. Please find our replies to your comments below. Your original comments are repeated in italics, our replies in normal font, and text passages which we included in the text are in bold.

*The study quantifies the contributions of emissions from the transport sector to tropospheric ozone and the hydroxyl radical (OH) using a global chemistry-climate model equipped with a source tagging method. The contributions are estimated for present-day level and several future scenarios. The novelty largely lies in the tagging techniques used to account for the non-linear source contribution to ozone and OH, as well as the analysis of different scenarios, which can provide new insights into controlling emissions from the transport sector. Overall, the study is well-designed and falls within the scope of ACP. I have a few suggestions.*

Reply: Thanks a lot for your positive comments and your suggestions. Please find below more detailed responses to your comments and changes in the manuscript.

*1.Introduction: The rationale for emphasizing emissions from the transport sector in this study needs to be clarified. Is it due to the large scale of emissions from the transport sector, or is it because these emissions are expected to undergo significant changes in the future and across different scenarios, offering a potential means to mitigate air pollution?*

Reply: Thanks for this point. Indeed, the rationale is a combination of both. We added the following explanation in the introduction:

**The emissions of the transport sector are an important source of ozone precursors and other species affecting climate and air quality. Due to various efforts in reducing the effect of the transport sector on climate and air quality, for example shifts towards electric vehicles, the emissions of the transport sector will likely undergo large changes in the future. When designing such mitigation measures for the transport sector these non-linear processes in atmospheric chemistry need to be considered.**

*2. While the EMAC model has been widely used, there is a need for an evaluation of the model results, particularly regarding ozone in the free troposphere, where it has a stronger radiative impact. Additionally, it is important to assess how well the EMAC model captures present-day OH levels and methane lifetime.*

Reply: Our model configuration has only minor updates (e.g. more recent model version, changed emissions) compared to the configuration in Jöckel et al. (2016) including a detailed evaluation. We compared some key species with the results of the *RC1SD-base-10a* model simulation as described by Jöckel et al. (2016).

Overall, our analysis shows that the magnitude of the differences between the results of the two simulations is as expected given the different emission inventories. Therefore, we conclude that the detailed evaluation presented by Jöckel et al. (2016) holds also for the simulations analysed in the present manuscript. In addition, we compared upper tropospheric/lower stratospheric ozone simulation results with the SWOOSH data-set. This comparison confirms the known positive bias of ozone in the troposphere. Moreover, we want to stress that our goal was not to use the 'best' available emissions inventory for present day, but to use the CMIP6 emissions inventory in their original form, since model results based on these emissions inventories have been used in many studies.

In the revised manuscript we added a new, rather short, subsection 'Evaluation' and included additional resources in the Supplement.

The added section reads:

**The EMAC model has been extensively evaluated in the past. Jöckel et al. (2016) present a detailed evaluation of various atmospheric variables, including tropospheric and stratospheric ozone. From these evaluations we know that EMAC has a positive bias of tropospheric ozone and a negative bias of carbon monoxide. Estimates of the methane lifetime simulated by EMAC are typically at the lower end of the range of values estimated by other models. However, multi-model inter-comparisons show that the biases compared to observational data of EMAC are within the range of those of comparable models (Naik et al., 2013; Stevenson et al., 2013; Voulgarakis et al., 2013; Young et al., 2013).**
**Given these extensive previous evaluation efforts, we reduce the evaluation of our model results to a minimum. In a first step we compare the ozone mixing ratios of the results from our PD simulation with the results of the _RC1SD-base-10a_ simulation discussed by Jöckel et al. (2016). The set-up of both simulations are very similar, despite changes of the emission inventories, small updates and bug-fixes in the model infrastructure, and the fact that we simulate more recent years. Ozone is larger by 2−4 nmol mol$^{-1}$ in PD compared to _RC1SD-base-10a_ in the extra-tropical lower and middle troposphere. In the extra-tropical free troposphere the difference between the two simulation is slightly larger, reaching up to 8 nmol mol$^{-1}$. In the tropical troposphere the difference range between -2−2 nmol mol$^{-1}$. Overall, the change is lower than 8 % with the largest increase in the Southern Hemisphere, dominated by the variability of the polar vortex. Figures of the comparison of ozone and of further trace gases are provided in the Supplement (see Supplement Sect. S10). From this analysis we conclude that the extensive evaluation presented by Jöckel et al. (2016) remains valid.**
**In addition, we compared the simulated ozone mixing ratios in the**

upper troposphere and lower stratosphere (UTLS) with Satellite measurements published as the Stratospheric Water and OzOne Satellite Homogenized dataset (SWOOSH) by Davis et al. (2016). The SWOOSH data are a homogenized, gridded, monthly-mean data set for ozone and water vapour based on several satellite data. For the considered period the data set is based on Aura MLS. We used the SWOOSH data in version 2.6 with a horizontal resolution of 2.5° and 31 vertical levels. Horizontally, the SWOOSH data are interpolated onto the slightly coarser EMAC grid, vertically the data are interpolated onto the much coarser SWOOSH grid similar as by Pletzer and Grewe (2024). The monthly-mean SWOOSH data are compared with monthly-mean data from the model, meaning that satellite data and model data are not co-located in space and time. Averaging Kernels of the Satellite are not considered, accordingly the satellite data can only be used for a qualitative evaluation. The evaluation is performed for the years 2013−2017.

Figure A1 shows the difference between the ozone mixing ratios of the PD simulation and the SWOOSH data. Overall, the inter-comparison confirms the known bias of simulated ozone, as discussed above, also in the upper troposphere. We would like to stress that the results can only be used for qualitative evaluation (i.e. confirming the ozone bias), as neither averaging kernels are used, nor are the data spatially and temporally co-located. Moreover, the number of considered years are very limited and we found that the magnitude and location of the peak of the upper tropospheric ozone bias strongly depends on the approach used for vertical remapping due to the limited vertical resolution of SWOOSH. For a detailed quantitative evaluation of UTLS ozone we refer to previous inter-comparisons for example with the IAGOS (in-situ measurements on board passenger aircraft) measurements presented by Jöckel et al. (2016); Pletzer et al. (2022); Cohen et al. (2024).

*3. Figure 10 is an excellent illustration of the non-linear nature of ozone chemistry and the higher ozone production efficiency from aviation emissions. Would it be feasible to perform a comparable calculation for the radiative efficiency from land transport and aviation? Can we expect that aviation-emitted NOx has a significantly higher radiative efficiency, as indicated by Wang et al. (2022)? Wang, H., et al. Global tropospheric ozone trends, attributions, and radiative impacts in 1995–2017: an integrated analysis using aircraft (IAGOS) observations, ozonesonde, and multi-decadal chemical model simulations, Atmos. Chem. Phys., 22, 13753–13782,*

Reply: We are not sure if we understand your comment correctly. We do analyse the radiative efficiency in Sect. 5.1 for both, the land transport and the aviation sectors. However, Wang et al. (2022) investigated ozone changes, whereas we are analysing contributions for a specific year. Hence a direct intercomparison is

difficult. Figure 11 shows that aviation radiative forcing (RF) is around 25-50% of the RF from land transport, whereas Figure 12 clearly indicates that the RF efficiency, e.g. for PD, is roughly 2.5 to 5 times larger than for land transport, which seems to be consistent with Wang et al. (2022) and also Dahlmann et al. (2011).

*4. Section 4: While the impacts of NOx and ozone on OH are discussed (Line 535), changes in CO and VOCs emissions also influence OH and methane lifetime, yet they are not addressed in this section. This discussion should be included.*

Reply: Of course, changes of emissions of CO, VOCs, etc. also affect the methane lifetime. We clarified our statement. The changed paragraph reads:

[revised manuscript text omitted]

---

## Author Comment (AC2)

Dear Michael Prather,

we thank you very much for your in-depth review of our manuscript egusphere-2024-324. Please find our replies to your comments below. Your original comments are repeated in italics and our replies in normal font.

*The open review process under ACP is a great opportunity to have a fair and public discussion of the core element of this paper: the concept of tagging of chemical species like O3 that has been developed by Volker Grewe and his colleagues. First, in terms of review, this paper does an excellent job of calculating the global impacts of three different SSP scenarios with the MESSY model. That alone wit a little more documentation of the current model is publishable.*

Reply: Thanks for the positive comment. According to the comments from referee #1 and referee #3 we added more documentation to the manuscript.

*Where I have a problem is with the tagging methodology. I had to re-read the core Mertens 2020 paper (Atmos. Chem. Phys., 20, 7843–7873) to try again to understand why one would want tagging versus sensitivity studies. At the time, that 2020 paper had some difficulty in convincing the reviewers of the usefulness of tagging for a chemical system in which there are many feedbacks (as for CH4 and O3). The Mertens Table 1 helped explain the difference between sensitivity and tagging, but it did not give me confidence in the usefulness of the tagging "attribution" value. I am not sure that 100% of the O3 in the troposphere must be attributed to something.*

Reply: First of all, we appreciate the positive feedback on our attempts to explain the differences in the methods in the respective Table 1 of Mertens et al. (2020). In addition to that, our point of view is as follows: all ozone in the atmosphere does have a source i.e., it is produced chemically by photolysis of oxygen and by photo-chemistry of other ozone precursors. That implies that 100 % of the calculated ozone has a specific source. The tagging approach decomposes ozone quantitatively, relating it to the shares of the different ozone precursor emission sectors and/or regions. Similar tagging approaches have been developed and used by various groups for a long time. Some examples are given here:

- Horowitz and Jacob (1999): $NO_x$ Tagging

- Lelieveld and Dentener (2000): Labelling technique for $NO_x$ and $O_3$.

- Emmons et al. (2010): Ozone tagging mechanism for MOZART

- Butler et al. (2018): TOAST 1.0 Ozone Tagging Mechanism

All of them are based on (partly) different assumptions (see for example detailed discussion by Butler et al., 2018, 2020), but all of them have the same goal: to explain 100 % of ozone for a given chemical state of the atmosphere. The latter is in contrast to the perturbation method, which targets on explaining **changes**

of ozone under perturbation of its precursors. These are two different, but complementary aspects.

*If this is a misreading, please let me know. My point of view is that given the indistinguishable nature of O3 molecules – from whichever source – labeling such a molecule is simply not useful. The essence of any calculation for policy options should be simply what happens if a policy is invoked. For that purpose, I can understand how sensitivity runs give the correct answer, but tagging may or may not.*

Reply: Here, we do not claim that the tagging method provides the only information for policy makers (see Mertens et al., 2020, Table 1), and we agree that this method is not primarily meant for this. However, there are also disadvantages of the perturbation method, since the timing of a sequence of policy measures might largely impact their environmental impact (Grewe et al., 2012). In addition, the perturbation method is not well suited to asses the share of one specific emission source to ozone by turning the emissions on/off because of the non-linerity of the ozone chemistryEmmons et al. (2010); Grewe et al. (2017); Mertens et al. (2018). Therefore, in our study we apply both methods, the source apportionment method to (scientifically) understand the share of different emission sectors in ozone for a given emission scenario. With the further sensitivity (perturbation) simulations, we assess if and how ozone and the ozone shares change, and what the implications for policy making are.

*As a lesson, one can look at the idea of labeling/tagging CH4. If one emits a Tg of CH4 and colors it uniquely, we find it decays with the lifetime timescale (e.g., 9 yr), but if we model CH4 as a whole, we find that 99.5% of that Tg perturbation decays with the perturbation time scale (e.g., 12 yr). Well, our colored/tagged CH4 does decay in 9 yr because the perturbation to OH is small, but the remaining atmospheric methane responds to the added Tg and alters the abundance of the untagged CH4, so as to make the overall perturbation decay in 12 yr. Thus, the colored/tagged CH4 does not represent the system behavior, nor even the attributable response to the emission. This example is one of the fundamentals of atmospheric chemistry that we had to develop to "explain" the seemingly nonsensical behavior of a small CH4 perturbation, and it is why a linear attribution that sums to 100% is troublesome to me. For O3, the situation is the same, but in opposite sense. With the ATom data (Deconstruction of tropospheric chemical reactivity using aircraft measurements: the Atmospheric Tomography Mission (ATom) data, Earth Syst. Sci. Data, 15, 3299–3349, doi:10.5194/essd-15-3299-2023), we showed that increases in O3 result in significant drops in production (log sensitivity = -0.4) in addition to increased loss (log sensitivity   1). Thus, an O3 perturbation reduces net P-L for all tropospheric O3 and the perturbation decays much faster than expected (opposite to CH4). We recently showed that the impact of the stratosphere-troposphere exchange (STE) flux of O3 was much less than expected because of these chemical feedbacks (2024. Lifetimes and timescales of tropospheric ozone. Elem. Sci.*

*Anth., 12: 1. doi: 10.1525/elementa.2023.00112).*

Reply: We happily agree to the sentence: "why a linear attribution that sums to 100% is troublesome to me". The point that bothers us is the thinking of a linear attribution. Here we use a highly non-linear technique, as also done e.g. in Emmons et al. (2010) and Butler et al. (2018), though differently. This non-linear decomposition is described in Grewe (2013). In Section 5 of that paper (Comparison of diagnostical methods) a simple non-linear differential equation is given. If we set alpha=1.5 in that equation, we can obtain a lifetime of 8 years and a perturbation lifetime of 12 years for that arbitrary species. Hence, any, perturbation decay then with a lifetime of 12 years, while the unperturbed lifetime is still 8 years. And we can analyze the transient lifetime of the perturbed situation analytically. We should not confuse these two approaches: An analysis of a current state and changes due to e.g. emission changes. We think the misunderstanding here arises from the thought of a linearisation that might be used in an attribution method, which is not the case here. We further think that in the comment the two approaches are indeed confused, which is indicated by the sentence *"... or even the attributable response to the emission .."*: We think that this might be the key to common misunderstandings: we never claimed that the "colored shares" provide any information about the systems "response characteristics"! Moreover, we do not apply any upscaling of perturbation results to 100%. And last, but not least, an "increase" *"... we showed that increases in O3 ..."* is the result of a "perturbation", but this is not what we do want to assess with the tagging method!

*In particular, the use of tagged O3S tracers for attributing the role of STE in tropospheric O3 is found to be mistaken because the simple, linear loss does not include the reduced production for O3 of tropospheric 'origin'. The idea that an O3 molecule has an origin is flawed. The O3S tagged tracer is typically 30-40% of tropospheric O3, but the perturbation to tropospheric O3 caused by the total STE O3 flux is only about 8%.*

Reply: Again, we do not claim that our ozone decomposition by tagging is any measure for ozone perturbations. With our tagging approach we do not tag processes or perturbations, but we attribute ozone shares to their precursor sources by taking into account all non-linearities (feedbacks and non-linear chemistry) between precursor emission and ozone production. Thus, O3S is not a measure for the perturbation that stratospheric ozone exerts on tropospheric ozone, but it is simply the share of ozone that has been produced originally in the stratosphere. The contribution of our tagged ozone from stratospheric origin (i.e., all ozone produced by photolysis of oxygen) is around 5 - 20 % in the lower and middle troposphere (see Grewe et al., 2017).

*The authors are very worried about the non-linear O3-NOx relationship, but that is exaggerated since most of the results here are far from the pollution centers where the NOx-limited vs VOC-limited issues are fought. One of the*

*Mertens 2020 reviews notes that "the response of ozone to perturbation of precursor emissions in remote regions has been shown to be approximately linear," which I believe is true. Production of O3 is almost linear in NOx over the oceans as found in ATom. What we have globally for O3 and CH4 are chemical feedbacks caused by the non-linearity of chemistry – specifically, the reaction of two species always has 2nd-order Taylor expansion terms that produce a Jacobian with off-diagonal elements, which give us timescales that differ from lifetimes and indirect greenhouse gases (i.e., NOx and CO alter the CH4 timescale).*

Reply: And just a side comment, we are not worried about non-linearities. It's the core of the fascinating atmospheric chemistry. Figure 10 clearly shows the non-linear behaviour, if emissions are reduced. See also the comment from referee#1.

*My review is not intended to prevent publication of this manuscript in ACP, but I would like to be able to understand how tagging helps us understand how to alter emissions to produce a better result in such a coupled world.*

Reply: This comment is very much appreciated and an excellent example of the online discussion forum. The short answer with which we think that we might be able to point at a misunderstanding is twofold:

1. The tagging mechanism is non-linear.

2. The tagging method does not provide information about the effect of altered (i.e., perturbed) emissions and we never claimed that it does.

**References**

Butler, T., Lupascu, A., Coates, J., and Zhu, S.: TOAST 1.0: Tropospheric Ozone Attribution of Sources with Tagging for CESM 1.2.2, Geosci. Model Dev., 11, 2825–2840, https://doi.org/10.5194/gmd-11-2825-2018, 2018.

Butler, T., Lupascu, A., and Nalam, A.: Attribution of ground-level ozone to anthropogenic and natural sources of nitrogen oxides and reactive carbon in a global chemical transport model, 20, 10 707–10 731, https://doi.org/10.5194/acp-20-10707-2020, 2020.

Emmons, L. K., Walters, S., Hess, P. G., Lamarque, J.-F., Pfister, G. G., Fillmore, D., Granier, C., Guenther, A., Kinnison, D., Laepple, T., Orlando, J., Tie, X., Tyndall, G., Wiedinmyer, C., Baughcum, S. L., and Kloster, S.: Description and evaluation of the Model for Ozone and Related chemical Tracers, version 4 (MOZART-4), Geosci. Model Dev., 3, 43–67, https://doi.org/10.5194/gmd-3-43-2010, 2010.

Grewe, V.: A generalized tagging method, Geosci. Model Dev., 6, 247–4253, https://doi.org/10.5194/gmdd-5-3311-2012, 2013.

Grewe, V., Dahlmann, K., Matthes, S., and Steinbrecht, W.: Attributing ozone to NOx emissions: Implications for climate mitigation measures, Atmos. Environ., 59, 102–107, https://doi.org/10.1016/j.atmosenv.2012.05.002, 2012.

Grewe, V., Tsati, E., Mertens, M., Frömming, C., and Jöckel, P.: Contribution of emissions to concentrations: the TAGGING 1.0 submodel based on the Modular Earth Submodel System (MESSy 2.52), Geosci. Model Dev., 10, 2615–2633, https://doi.org/10.5194/gmd-10-2615-2017, 2017.

Horowitz, L. W. and Jacob, D. J.: Global impact of fossil fuel combustion on atmospheric NO, Journal of Geophysical Research: Atmospheres, 104, 23 823–23 840, https://doi.org/https://doi.org/10.1029/1999JD900205, 1999.

Lelieveld, J. and Dentener, F. J.: What controls tropospheric ozone?, J. Geophys. Res. Atmos., 105, 3531–3551, https://doi.org/10.1029/1999JD901011, 2000.

Mertens, M., Grewe, V., Rieger, V. S., and Jöckel, P.: Revisiting the contribution of land transport and shipping emissions to tropospheric ozone, Atmos. Chem. Phys., 18, 5567–5588, https://doi.org/10.5194/acp-18-5567-2018, 2018.

Mertens, M., Kerkweg, A., Grewe, V., Jöckel, P., and Sausen, R.: Attributing ozone and its precursors to land transport emissions in Europe and Germany, 20, 7843–7873, https://doi.org/10.5194/acp-20-7843-2020, 2020.

---

## Author Comment (AC3)

Dear referee#3,
we thank you very much for your in-depth review of our manuscript egusphere-2024-324. Please find our replies to your comments below. Your original comments are repeated in italics, our replies in normal font, and text passages which we included in the text are in bold.

*General comments: The manuscript offers a comprehensive assessment of transport emissions' impacts on ozone and the hydroxyl radical (OH) across different transport sectors using the EMAC Chemistry-Climate model's simulations. The study uses an innovative approach to quantify contributions to OH for the years 2015 and projections for 2050, under various SSPs. The analysis extends to ozone radiative forcing and methane lifetime reductions, highlighting the manuscript's value to environmental policy and planning stakeholders. The manuscript is well written and provides an extensive analysis of the impacts arising from various emission scenarios. However, there are several areas where further development could enhance the study's robustness. After addressing the suggestions outlined below, the manuscript should be considered for publication due to its valuable contribution to the field.*

Reply: Thanks a lot for your overall positive comments on our manuscript. We revised the manuscript based on your suggestions and the comments from the other referees. Please find our detailed comments and changes below.

*Model Evaluation: The manuscript would greatly benefit from a dedicated validation section. Such a section should detail the model's proficiency in simulating the chemical environment and meteorology for the base year of 2015. This could include comparisons of model outputs with observed data or results from prior studies to establish the model's skills*

Reply: Thanks a lot for your suggestion. Referee#1 raised a similar point. Therefore, we added a short model evaluation with additional details in the Supplement.
Generally, our model configuration has only minor updates (e.g. more recent model version, changed emissions) compared to the configuration in Jöckel et al. (2016) including a detailed evaluation. We compared some key species with the results of the *RC1SD-base-10a* model simulation as described by Jöckel et al. (2016). Overall, our analysis shows that the magnitude of the differences between the results of the two simulations is as expected given the different emission inventories. Therefore, we conclude that the detailed evaluation presented by Jöckel et al. (2016) holds also for the simulations analysed in the present manuscript. In addition, we compared upper tropospheric/lower stratospheric ozone simulation results with the SWOOSH data-set. This comparison confirms the known positive bias of ozone in the troposphere. Moreover, we want to stress that our goal was not to use the 'best' available emissions inventory for present day, but to use the CMIP6 emissions inventory in their original form, since model results based on these emissions inventories have been used in many

studies.
The added subsection reads:

The EMAC model has been extensively evaluated in the past. Jöckel et al. (2016) present a detailed evaluation of various atmospheric variables, including tropospheric and stratospheric ozone. From these evaluations we know that EMAC has a positive bias of tropospheric ozone and a negative bias of carbon monoxide. Estimates of the methane lifetime simulated by EMAC are typically at the lower end of the range of values estimated by other models. However, multi-model inter-comparisons show that the biases compared to observational data of EMAC are within the range of those of comparable models (Naik et al., 2013; Stevenson et al., 2013; Voulgarakis et al., 2013; Young et al., 2013).

Given these extensive previous evaluation efforts, we reduce the evaluation of our model results to a minimum. In a first step we compare the ozone mixing ratios of the results from our PD simulation with the results of the *RC1SD-base-10a* simulation discussed by Jöckel et al. (2016). The set-up of both simulations are very similar, despite changes of the emission inventories, small updates and bug-fixes in the model infrastructure, and the fact that we simulate more recent years. Ozone is larger by $2-4$ nmol mol$^{-1}$ in PD compared to *RC1SD-base-10a* in the extra-tropical lower and middle troposphere. In the extra-tropical free troposphere the difference between the two simulation is slightly larger, reaching up to 8 nmol mol$^{-1}$. In the tropical troposphere the difference range between -2$-2$ nmol mol$^{-1}$. Overall, the change is lower than 8 % with the largest increase in the Southern Hemisphere, dominated by the variability of the polar vortex. Figures of the comparison of ozone and of further trace gases are provided in the Supplement (see Supplement Sect. S10). From this analysis we conclude that the extensive evaluation presented by Jöckel et al. (2016) remains valid.

In addition, we compared the simulated ozone mixing ratios in the upper troposphere and lower stratosphere (UTLS) with Satellite measurements published as the Stratospheric Water and OzOne Satellite Homogenized dataset (SWOOSH) by Davis et al. (2016). The SWOOSH data are a homogenized, gridded, monthly-mean data set for ozone and water vapour based on several satellite data. For the considered period the data set is based on Aura MLS. We used the SWOOSH data in version 2.6 with a horizontal resolution of 2.5° and 31 vertical levels. Horizontally, the SWOOSH data are interpolated onto the slightly coarser EMAC grid, vertically the data are interpolated onto the much coarser SWOOSH grid similar as by Pletzer and Grewe (2024). The monthly-mean SWOOSH data are compared with monthly-mean data from the model, meaning that satellite data and model data are not co-located in space and time. Averaging Kernels

of the Satellite are not considered, accordingly the satellite data can only be used for a qualitative evaluation. The evaluation is performed for the years 2013−2017.

Figure A1 shows the difference between the ozone mixing ratios of the PD simulation and the SWOOSH data. Overall, the inter-comparison confirms the known bias of simulated ozone, as discussed above, also in the upper troposphere. We would like to stress that the results can only be used for qualitative evaluation (i.e. confirming the ozone bias), as neither averaging kernels are used, nor are the data spatially and temporally co-located. Moreover, the number of considered years are very limited and we found that the magnitude and location of the peak of the upper tropospheric ozone bias strongly depends on the approach used for vertical remapping due to the limited vertical resolution of SWOOSH. For a detailed quantitative evaluation of UTLS ozone we refer to previous inter-comparisons for example with the IAGOS (in-situ measurements on board passenger aircraft) measurements presented by Jöckel et al. (2016); Pletzer et al. (2022); Cohen et al. (2024).

*Model Description: The description of the EMAC model setup, including its chemical mechanisms, is thorough. Nonetheless, the manuscript would benefit from additional details on the model's parameterizations, particularly those influencing ozone and other chemical species. This should encompass radiation, deposition, and boundary layer schemes, as well as the land surface model used. A clarification on whether the simulations incorporate direct radiation feedback would be pertinent.*

Reply: Thanks for this suggestion. As stated in the manuscript the model set-up is very similar to the set-up of (Jöckel et al., 2016). Therefore, we wanted to keep repetitions to a minimum, but of course the information in the manuscript should reflect the most important details. However, we added some more details on the mentioned processes. In Sect. 2 we described the 'QCTM' mode of EMAC which is applied in all simulations. In this operation mode the same (prescribed) climatologies from previous simulations are used for all radiatively active trace gases. With this approach we achieve identical model dynamics in all simulations. This approach is very important to be able to quantify even small perturbations (see Deckert et al., 2011). These small perturbations might not be detectable in a statistically robust way or require very long integration times.

To make this more clear in the manuscript we added some more details on the description of the QCTM mode.

In the description of the RF calculation: **It is important to note that neither the radiative fluxes from $O_3$ nor the ozone contributions (e.g. $O_3^{\mathrm{SHP}}$) feed back back onto the dynamics. Instead, prescribed cli-**

matologies are used for the forcing of the dynamics (as described in Sect. 2.4).

In the description of the QCTM mode: In this mode, mixing ratios of the radiatively active trace gases are prescribed for the radiation calculations. **This means that in each simulation the same radiative forcings by the prescribed mixing ratios are considered.**

As stated in the manuscript the model set-up is very similar to the set-ups described and evaluated by Jöckel et al. (2016) including also detailed description of the considered processes (see als Jöckel et al., 2010). We expanded our short description with details on the dry deposition scheme, calculation of phtolysis rates, the radiation scheme, the boundary layer and the land-surface model. The added points are:

- Heterogeneous reactions in the stratosphere (submodel MSBM, Jöckel et al., 2010) as well as aqueous phase chemistry and scavenging (submodel SCAV, Tost et al., 2006) are included. **Photolysis rates are calculated using JVAL (Sander et al., 2014).**

- **Dry deposition is considered via the submodel DDEP (described as DRYDEP by Kerkweg et al., 2006). It is based on the big-leaf approach by Wesely and Hicks (2000).**

- **The radiation is largely based on the original radiation scheme from ECHAM5 (Roeckner et al., 2003), but restructured and expanded with additional features such as multiple diagnostic calls as described by Dietmüller et al. (2016).**

- **The land surface model and the boundary layer implementation are modularized versions (see also Jöckel et al., 2016) of the original implementations of ECHAM5 described in detail by (Roeckner et al., 2003).**

*Introduction and Methods: Structural Suggestions: Consider relocating parts of the limitations and uncertainties discussion, currently in line 610, to the introduction or methodology sections. This would help set the reader's expectations early in the manuscript.*

Reply: Thanks for the suggestion. We would like to keep the structure as it is, because we expanded/restructured the discussion a lot based on the reviews/community comments. Nevertheless, we did add some information about the limitations in the introduction. The changed paragraph reads:

We performed simulations for 2015 and for the three considered SSPs. **Each simulation covers five years and simulates the same present-day meteorology. Accordingly, the influence of climate change on atmospheric**

**composition is not considered, but the effect on the results are discussed in detail.**

*Climate Change Impact: Incorporating a discussion on the potential changes and impacts of climate change on atmospheric chemistry and transport patterns is recommended. This includes a thorough consideration of radiation feedbacks and their prospective effects on future climate change scenarios. For example, changes in surface and atmospheric temperatures can profoundly influence ozone chemistry; as temperatures increase, so do evaporation rates, which lead to a higher concentration of water vapor in the atmosphere, potentially affecting OH and ozone levels. Additionally, changes in cloud cover can alter photolysis rates, thereby impacting ozone formation and destruction*

Reply: Thanks for the suggestions. The community comment from Owen Cooper raised similar concerns. We restructured the discussion and expanded our discussion on the effect of climate change on ozone. We want to stress that the effects of climate change are not considered in these simulations and we can't asses the effects of radiation feedbacks from our model results by design. This is due to the use of the 'QCTM' mode. For further studies it is very important to also include the effects of climate change on atmospheric chemistry. This would, however, need a different model set-up with much more expansive and much longer time-slice and transient simulations due to the large signal-to-noise ratio.

The changed discussion reads: **Effects of climate change**
The signals of emissions from specific regions or specific emission sources (such as e.g. aviation) are small. To quantify these signals, we apply the QCTM mode (see Sect. 2) in which chemistry and dynamics are decoupled. Accordingly, the dynamics (and therefore the climatic state) is identical in every model simulation. This approach of applying present day dynamics for future emission scenarios is commonly used when the effects of certain emission changes or sources on the atmospheric chemistry are investigated on the global and regional scale (e.g., Eyring et al., 2007; Hoor et al., 2009; Hodnebrog et al., 2012; Righi et al., 2015; Matthias et al., 2016).
Due to this approach, however, **our model simulations do not consider changes of meteorology and climate between 2015 and 2050. Accordingly**, emissions which are based on meteorological conditions (e.g. biogenic emissions, lightning-$NO_x$) are identical in all simulations. With climate change, these emissions are likely to increase (von Schneidemesser et al., 2015). This increase could alter the contributions of the anthropogenic emissions, for instance increased biogenic VOC emissions may affect the ozone production efficiency, while increased lightning-$NO_x$ in the upper troposphere may compete with $NO_x$ emissions from the aviation sector.
**Moreover, increased biogenic emissions and changed atmospheric conditions (e.g. increased temperature and it's effects on kinetics) likely lead to an increase of ozone near highly polluted regions (knows as**

'climate-penalty', Zanis et al., 2022). In addition, climate change likely leads to an decrease of ozone in remote regions due to the increase of water vapour (known as 'climate-benefit', Zanis et al., 2022). In addition, during periods of droughts and heat-waves, reduced ozone deposition to vegetation could increase ground-level ozone (Lin et al., 2020).

Altogether, this could affect also the contributions of the traffic emissions. A reduced life-time of ozone, especially over the oceans, would likely lead to a reduction of ozone attributable to shipping emissions. Also long-range transport, especially the source-receptor relationships, might be affected by changes of the ozone lifetime. At the same time, the increase of ozone in polluted regions in a changing climate could affect ozone contributions especially from land transport emissions.

Koffi et al. (2010) considered the effects of climate change on the ozone effects of transport emissions applying a 5 % emissions reduction. Globally, they report a small decrease of the ozone changes caused by transport emissions due to climate change, but with strongly varying regional patterns. **The effect of climate change on ozone contributions (i.e. applying a tagging approach) needs to be analysed in follow-up studies.**

*Land Surface Model Considerations: It is important to discuss the implications of land surface model choices within the simulations. Soil moisture variability, alterations in land use, and vegetation cover driven by climate scenarios play an important role in the soil's chemical processes and the land's overall energy budget. For instance, soil moisture dependent on the chosen climate scenario affects soil chemistry, influencing how land surface models simulate these processes. Similarly, changes in land use and vegetation cover have the potential to modify the absorption, reflection, and emission of radiant energy at the land surface. Moreover, the type of vegetation and temperature changes can affect the deposition of chemical species and their uptake by plants.*

Reply: We agree that these aspects also affect ozone. As stated in the reply above effects of climate, land use etc. are not considered in the present study. We added a short note about these effects in the overall discussion. This part reads:

**Similar to changes of the climate, changes in land-use also affect ozone (e.g. Wu et al., 2012; Wang et al., 2020). The effects of land-use change on ozone are due to various processes, such as changes of biogenic emissions, effects on dry deposition, and changes of temperature (e.g., by effects on radiation and evapo-transpiration). Land-use change, and the corresponding effects, also heavily depend on the considered scenario (Popp et al., 2017). These effects are not considered to isloate the effects of the emissions changes only.**

*Sensitivity Analysis: Could a sensitivity analysis be performed to evaluate how various model assumptions, such as chemical reaction rates and deposition processes in a changing climate, might affect the outcomes? This would contribute to understanding the study's conclusions' robustness.*

Reply: Thanks for the suggestions. As stated above our model set-up currently does not consider the effects of climate change by design. Therefore, the current model set-up is not suitable to asses questions such as "chemical reaction rates and deposition processes in a changing climate, might affect the outcomes?". This would require a different model set-up which is out of scope of the present study. However, as stated above, this would be the next step for follow up studies.

**References**

[revised manuscript text omitted]

---

## Author Comment (AC4)

Dear Owen Cooper,
we thank you very much for your detailed community comment of our manuscript egusphere-2024-324. Please find our replies to your comments below. Your original comments are repeated in italics, our replies in normal font, and text passages which we included in the text are in bold.

*This paper provides a thorough analysis of the impact of transport emissions on present-day and future (2050) ozone based on three different SSPs. This is a complex endeavor requiring a wide range of tagged tracer runs and sensitivity tests, and it's not possible to consider every situation and account for every competing process (e.g. emissions, climate change, non-linearity).*

Reply: Thanks a lot for your detailed comment and your overall positive feedback. We incorporated your suggestions, which helped to improve the revised manuscript strongly.

*The authors are of course aware of this challenge and provide some extensive discussion in Section 7. I think this section would benefit from some further discussion regarding SSP3-7.0 and the expected impacts of climate change and increasing methane concentrations, as assessed in Chapter 6 of IPCC AR6 WG-I (Szopa et al., 2021).*

Reply: Thanks for your suggestion. We changed the discussions section and especially included more information on methane and climate change aspects (see below for detailed comments).

*Figure 6.4 in Szopa et al. shows an increase of the tropospheric ozone burden of roughly 10% from 2014 to 2050, based on SSP3-7.0, and much of this increase is due to projected increases in methane. Figure 6.20 in Szopa et al. indicated average ozone increases across South Asia of 8-10 ppb by 2050, under SSP3-7.0. These ozone increases seem to be much larger than your projected increases, as shown in your Figure 2. Part of this discrepancy could be due to differences in methane concentrations, as you discussed in Section 7.*

Reply: Our model results for SSP3-7.0 show an increase over Southern Asia of 4–8 $\mathrm{nmol\,mol^{-1}}$. We agree that a very likely reason for this discrepancy is the methane effect (either because of fixed methane lower boundary conditions or differences in the methane life-time). We performed our methane sensitivity simulation only for the SSP2-4.5 projection in 2050. Here, we find an increase of ozone over Asia in the range of $1-2$ $\mathrm{nmol\,mol^{-1}}$ at ground-level due to the increased methane levels (see Figure 1).

We added the following note in Sect. 3.1:

**However, the magnitude of the ozone change differs, especially our increase of ozone in SSP3.7-0 is lower as shown by Turnock et al.**

[Figure]

Figure 1: Absolute difference of ground-level ozone (in $\mathrm{nmol\,mol^{-1}}$) between SSP2-4.5 and the sensitivity simulation including increased methane lower boundary conditions.

**(2020). Moreover, our results for the SSP3-7.0 does not show the strong decrease of ozone over the oceans as discussed by Zanis et al. (2022). Both differences can be expected, as we keep the methane lower boundary condition to present-day values, and because we do not include the effects of changing meteorology and climate and therefore also have constant water vapour concentrations in all simulations (see Sect. 7 for a detailed discussion).**

*But another likely explanation is the ozone climate penalty that impacts boundary layer ozone, as discussed by Zanis et al. 2022. Your paper does not mention the climate penalty and I think that it deserves some discussion. Another important finding of IPCC AR6 and Zanis et al. (2022) is that a warmer climate will be more humid, especially in the boundary layer, which will lead to a reduction of ozone lifetime in remote regions, such as over the oceans. .*

Reply: Indeed, we need to address these points in more detail (see also comments from referee #3). The modified paragraph in the discussion reads:

Due to this approach, however, our model simulation do not consider changes in meteorology and climate between 2015 and 2050. Accordingly, emissions which are based on meteorological conditions (e.g. biogenic emissions, lightning-$NO_x$) are identical in all simulations. With climate change, these emissions are likely

to increase (von Schneidemesser et al., 2015). This increase could alter the contributions of the anthropogenic emissions, for instance increased biogenic VOC emissions may affect the ozone production efficiency, while increased lightning-$NO_x$ in the upper troposphere may compete with $NO_x$ emissions from the aviation sector.

**Moreover, increased biogenic emissions and changed atmospheric conditions (e.g. increased temperature and it's effects on kinetics) likely lead to an increase of ozone near highly polluted regions (knows as 'climate-penalty', Zanis et al., 2022). In addition, climate change likely leads to an decrease of ozone in remote regions due to the increase of water vapour (known as 'climate-benefit', Zanis et al., 2022). In addition, during periods of droughts and heat-waves, reduced ozone deposition to vegetation could increase ground-level ozone (Lin et al., 2020). Altogether, this could affect also the contributions of the traffic emissions. A reduced life-time of ozone, especially over the oceans, would likely lead to a reduction of ozone attributable to shipping emissions. Also long-range transport, especially the source-receptor relationships, might be affected by changes of the ozone lifetime. At the same time, the increase of ozone in polluted regions in a changing climate could affect ozone contributions especially from land transport emissions. Koffi et al. (2010) considered the effects of climate change on the ozone effects of transport emissions applying a 5 % emissions reduction (i.e. with the perturbation approach). Globally, they report a small decrease of the ozone changes caused by transport emissions due to climate change, but with strongly varying regional patterns. The effect of climate change on ozone contributions (i.e. applying a tagging approach) needs to be analysed in follow-up studies.**

*Your Figure 2 does not show a consistent reduction of ozone across the oceans under SSP3-7.0, probably because you use the same meteorology in 2015 and 2050; some discussion of this phenomenon would also be helpful.*

Reply: We fully agree with your analysis. Due to the same meteorology in all simulations, water vapour is identical in all simulations; i.e. we only consider the change of ozone due the changes of the ozone precursor emissions (despite methane). We added a short note on this in the discussion of the figure. The changed text reads:

In most regions the decrease is in the range of $10-15$ nmol mol$^{-1}$, and exceeds $20$ nmol mol$^{-1}$ on the Arabian Peninsula. The overall changes of ground-level ozone for the three projections and regional features, such as the strong increase of ozone over Asia in SSP3-7.0, are in agreement with the analyses of CMIP6 simulation results by Turnock et al. (2020). **However, the magnitude of the ozone change differs, especially our increase of ozone in SSP3.7-0 is lower as shown by Turnock et al. (2020). Moreover, our results for the**

**SSP3-7.0 does not show the strong decrease of ozone over the oceans as discussed by Zanis et al. (2022). Both differences can be expected, as we keep the methane lower boundary condition to present-day values, and because we do not include the effects of changing meteorology and climate and therefore also have constant water vapour concentrations in all simulations (see Sect. 7 for a detailed discussion).**

Moreover, we added a further note during at the end of our discussion of the influence of the fixed methane levels:

**Especially when considering our results of the SSP3-7.0 this effect should be kept in mind, because SSP3-7.0 shows even larger methane levels compared to SSP2-4.5.  .**

And a further note in the conclusion:

**Especially for the results of SSP3 it should kept in mind that we apply present day methane-levels in all simulations. Applying the methane levels for SSP3 in 2050 likely leads to even larger ozone increases, but the responses of the different emission sectors on the methane increase are very complex and require further investigations in follow up studies.**

*Figure 5. Given that SSP1-1.9 has strongly decreasing transport emissions in all regions, I am surprised that none of the regional reductions produces ozone reductions in downwind regions. Why are there no ozone reductions in the receptor regions?*

Reply: We are not sure whether we understand your comment correctly, or if this is simply a misunderstanding. The figure shows the absolute contribution of $O_3^{tra}$, which is always positive. So we don't expect to have negative values. However, if we plot the difference compared to PD (i.e. SSP1-1.9 minus PD) the values get negative showing that reductions exist (in agreement with Figs. 2 − 4, see also Fig. 2 in the reply which we also added to the revised supplement). Moreover, thanks to your comment we realized that the axis label for the color bar was wrong. We changed it from $O_3$ to $O_3^{tra}$.

*Figure 7. If the future scenarios included climate change, with more humidity in the boundary layer and therefore a shorter ozone lifetime, would the ozone reductions due to shipping emissions reductions be even more pronounced?*

Reply: This analysis seems plausible. We added a short discussion on this in the discussion Section. Yet, it remains to be tested whether counteracting effects on the tagged tracers exist. We added the following text:

**If climate-change would be considered in addition, the ozone contribution from shipping emissions could be reduced even more strongly in**

[Figure]

Figure 2: Source receptor analysis of the absolute contribution of land transport emissions to ground-level ozone (in nmol mol$^{-1}$). The values are mean values over 5 years and area weighted over the receptor regions. Exact definitions of the receptor regions are given in Sect. S9.1 in the Supplement. PD shows the absolute contributions for PD, all other panels show the difference of the absolute contributions compared to PD (e.g. SSP2-4.5 minus PD)

**the future, given the likely reduction of ozone over the oceans due to increasing humidity (Zanis et al., 2022, see also disucssion in Sect. 7).**

*Section 4.4 A recent paper by Wang et al. (2022) indicates that the impact of aviation on the global tropospheric ozone burden is greater than suggested by previous studies. How does your analysis compare to that of Wang et al.?*

Reply: This study was also mentioned by referee#1. It is difficult to compare our results directly to the results of Wang et al. (2022), because they calculate impacts on ozone levels from 1995 - 2017, while we calculate contributions at present day. However, as mentioned also in the reply to referee#1 our results in Fig. 10 and in Section 5.1. are in general agreement with Wang et al. (2022) and previous studies such as Dahlmann et al. (2011), indicating that aviation emissions are much more efficient in forming ozone compared to e.g. land transport emissions. Therefore, changes in aviation emissions can have stronger effects on tropospheric ozone compared to e.g. ground-level emissions. We added the study of Wang et al. (2022) accordingly in our manuscript.

*Minor Comments: Figure S4. There is hardly any difference in surface ozone*

*between PD and SSP3-7, which is surprising. SSP3-7 is projected to have an increase in the tropospheric ozone burden, especially in the free troposphere. This should mean that ozone at high elevations sites (Greenland, the western USA, Tibetan Plateau, the Andes, Antarctica) should be higher under SSP3-7, but they appear to be almost the same. Is this due to your 2015 and 2050 simulations having the same methane concentrations, instead of higher methane in 2050?*

Reply: We double checked the figure and compared it with Fig. 2 in the manuscript. The figures are consistent, but we agree with your comment that the lack of increasing methane levels are likely to be one of the reasons. We added this point in the manuscript in the same part where we addressed the point with the missing decrease of ozone over the oceans (see your comment above). The changed text is:

**However, the magnitude of the ozone change differs, especially our increase of ozone in SSP3.7-0 is lower as shown by Turnock et al. (2020). Moreover, our results for the SSP3-7.0 does not show the strong decrease of ozone over the oceans as discussed by Zanis et al. (2022). Both differences can be expected, as we keep the methane lower boundary condition to present-day values, and because we do not include the effects of changing meteorology and climate and therefore also have constant water vapour concentrations in all simulations (see Sect. 7 for a detailed discussion).**

*Line 622 When considering the impact of climate change on ozone, a relevant study is Lin et al. 2020, who show that drought and heat waves can limit ozone deposition to vegetation.*

Reply: Thanks for the additional reference/point. This is added!

*Line 410 When discussing ozone non-linearity, two relevant studies are Wu et al. (2009) and Wild et al. (2012). Similarly, when discussing differences in ozone production efficiency among regions, the study by Zhang et al. (2016) is very important as it demonstrated that ozone production efficiency is much greater in tropical regions than at northern mid-latitudes*

Reply: Thanks for the additional reference. They are now included!

**References**

Dahlmann, K., Grewe, V., Ponater, M., and Matthes, S.: Quantifying the contributions of individual NOx sources to the trend in ozone radiative forcing, Atmos. Environ., 45, 2860–2868, https://doi.org/http://dx.doi.org/10.1016/j.atmosenv.2011.02.071, 2011.

Koffi, B., Szopa, S., Cozic, A., Hauglustaine, D., and van Velthoven, P.: Present and future impact of aircraft, road traffic and shipping emissions on global tropospheric ozone, Atmospheric Chemistry and Physics, 10, 11 681–11 705, https://doi.org/10.5194/acp-10-11681-2010, 2010.

Lin, M., Horowitz, L. W., Xie, Y., Paulot, F., Malyshev, S., Shevliakova, E., Finco, A., Gerosa, G., Kubistin, D., and Pilegaard, K.: Vegetation feedbacks during drought exacerbate ozone air pollution extremes in Europe, Nature Climate Change, 10, 444–451, https://doi.org/10.1038/s41558-020-0743-y, 2020.

Turnock, S. T., Allen, R. J., Andrews, M., Bauer, S. E., Deushi, M., Emmons, L., Good, P., Horowitz, L., John, J. G., Michou, M., Nabat, P., Naik, V., Neubauer, D., O'Connor, F. M., Olivié, D., Oshima, N., Schulz, M., Sellar, A., Shim, S., Takemura, T., Tilmes, S., Tsigaridis, K., Wu, T., and Zhang, J.: Historical and future changes in air pollutants from CMIP6 models, 20, 14 547–14 579, https://doi.org/10.5194/acp-20-14547-2020, 2020.

von Schneidemesser, E., Monks, P. S., Allan, J. D., Bruhwiler, L., Forster, P., Fowler, D., Lauer, A., Morgan, W. T., Paasonen, P., Righi, M., Sindelarova, K., and Sutton, M. A.: Chemistry and the Linkages between Air Quality and Climate Change, Chemical Reviews, 115, 3856–3897, https://doi.org/10.1021/acs.chemrev.5b00089, pMID: 25926133, 2015.

Wang, H., Lu, X., Jacob, D. J., Cooper, O. R., Chang, K.-L., Li, K., Gao, M., Liu, Y., Sheng, B., Wu, K., Wu, T., Zhang, J., Sauvage, B., Nédélec, P., Blot, R., and Fan, S.: Global tropospheric ozone trends, attributions, and radiative impacts in 1995–2017: an integrated analysis using aircraft (IAGOS) observations, ozonesonde, and multi-decadal chemical model simulations, Atmospheric Chemistry and Physics, 22, 13 753–13 782, https://doi.org/10.5194/acp-22-13753-2022, 2022.

Zanis, P., Akritidis, D., Turnock, S., Naik, V., Szopa, S., Georgoulias, A. K., Bauer, S. E., Deushi, M., Horowitz, L. W., Keeble, J., Sager, P. L., O'Connor, F. M., Oshima, N., Tsigaridis, K., and van Noije, T.: Climate change penalty and benefit on surface ozone: a global perspective based on CMIP6 earth system models, Environmental Research Letters, 17, 024 014, https://doi.org/10.1088/1748-9326/ac4a34, 2022.